# Dendrite intercalation between epidermal cells tunes nociceptor sensitivity to mechanical stimuli in *Drosophila* larvae

**Kory P. Luedke, Jiro Yoshino**, **Chang Yin, Nan Jiang, Jessica M. Huang, Kevin Huynh**, **Jay Z. Parrish** *

Department of Biology, University of Washington, Seattle, Washington State, United States of America

* jzp2@uw.edu

## Abstract

An animal's skin provides a first point of contact with the sensory environment, including noxious cues that elicit protective behavioral responses. Nociceptive somatosensory neurons densely innervate and intimately interact with epidermal cells to receive these cues, however the mechanisms by which epidermal interactions shape processing of noxious inputs is still poorly understood. Here, we identify a role for dendrite intercalation between epidermal cells in tuning sensitivity of *Drosophila* larvae to noxious mechanical stimuli. In wild-type larvae, dendrites of nociceptive class IV da neurons intercalate between epidermal cells at apodemes, which function as body wall muscle attachment sites, but not at other sites in the epidermis. From a genetic screen we identified *miR-14* as a regulator of dendrite positioning in the epidermis: *miR-14* is expressed broadly in the epidermis but not in apodemes, and *miR-14* inactivation leads to excessive apical dendrite intercalation between epidermal cells. We found that *miR-14* regulates expression and distribution of the epidermal Innexins ogre and Inx2 and that these epidermal gap junction proteins restrict epidermal dendrite intercalation. Finally, we found that altering the extent of epidermal dendrite intercalation had corresponding effects on nociception: increasing epidermal intercalation sensitized larvae to noxious mechanical inputs and increased mechanically evoked calcium responses in nociceptive neurons, whereas reducing epidermal dendrite intercalation had the opposite effects. Altogether, these studies identify epidermal dendrite intercalation as a mechanism for mechanical coupling of nociceptive neurons to the epidermis, with nociceptive sensitivity tuned by the extent of intercalation.

**Data Availability Statement:** Raw data and details of data analysis are provided in the supplemental materials. RNA-seq data is available in the Gene

## Author summary

Our skin provides a first point of contact for a variety of sensory inputs, including noxious cues that elicit pain. Although specialized interactions between skin cells and sensory neurons are known to shape responses to a variety of mechanosensory stimuli including gentle touch and vibration, interactions with skin cells that shape responses to painful mechanical inputs are less well defined. Using the fruit fly *Drosophila melanogaster* as a

Expression Omnibus (GEO) under accession number GSE262604.

**Funding:** This work was supported by funding from the National Institutes of Health (NINDS R01 NS076614 and NINDS R21NS125795 to J.Z.P; NIGMS T32 GM007108-40 to K.P.L.), awards from the Weill Neurohub and the Scan Design Foundation, a JSPS long-term fellowship and startup funds from UW to J.Z.P. The work was additionally supported by funding from the Robin M. Harris award and WRF-Hall award to K.P.L. from the UW Dept. of Biology. Fly stocks obtained from the Bloomington Drosophila Stock Center (NIHP40OD018537) and antibodies obtained from the Developmental Studies Hybridoma bank, created by the NICHD of the NIH and maintained at The University of Iowa, were used in this study. The funders had no role in study design, data collection and analysis, decision to publish, or preparation of the manuscript. The following authors received salary support from the funders: J.Z.P. (NINDS R01 NS076614, NINDS R21NS125795, Weill Neurohub, and Scan Design Foundation); K.P.L. (NIGMS T32 GM007108-40, NINDS R01 NS076614, and the WRF-Hall award); J.Y. (NINDS R01 NS076614 and the Weill Neurohub); C.Y. (NINDS R01 NS076614); J.M.H. (NINDS R01 NS076614 and NINDS R21NS125795).

**Competing interests:** The authors declare no competing interests.

model system, we demonstrate that the pattern of epidermal innervation, specifically the extent of dendrite intercalation between epidermal cells, tunes the animal's sensitivity to noxious mechanical stimuli. Similar mechanisms may regulate sensitivity to painful mechanical inputs in both pathological and physiological states in vertebrates.

## Introduction

Somatosensory neurons (SSNs) shape our experience of the world, allowing for perception and discrimination of noxious (painful) inputs, touch, pressure, and movement. Among these, nociception is of particular interest both because it is a deeply conserved function of nervous systems and because of the adverse effect of pain on quality of life. Current estimates suggest that one in three individuals will suffer from chronic pain [1], with hypersensitivity to mechanosensory stimuli among the most prevalent complaints in the clinic [2]. Why do we have so much difficulty dealing with (and treating) pain? First, painful stimuli come in many forms, including noxious touch, heat, and chemicals, and our understanding of how these stimuli, in particular mechanosensory inputs, activate nociceptive SSNs is still limited. Second, pain is subjective, and individuals experience pain differently, the combined result of experience, genetic, and cultural factors that shape perception and responses to pain [3]. Third, although epidermal cells provide the first point of contact for sensory stimuli, the molecular mechanisms by which epidermal cells shape responses to noxious stimuli are largely unknown. This gap in our knowledge is particularly significant given the prevalence of pathological skin conditions associated with debilitating pain.

Several lines of evidence suggest that epidermal cells are key regulators of nociception. First, peripheral arbors of some nociceptive neurons are ensheathed in mesaxon-like structures by epidermal cells [4], and this epidermal ensheathment influences sensitivity to noxious mechanical stimuli in *Drosophila* [5]. Second, epidermal cells release a variety of compounds that can modulate nociceptive SSN function, notably including ATP, cytokines, and prostaglandins [6–8]. Third, epidermal cells express a variety of sensory channels notably including TRPV3 and the calcium release activated calcium (CRAC) channel ORAI, both of which contribute to thermal responses in mice [9, 10]. Nociceptive functions for epidermal channels are less well-defined, but UVB activation of epidermal TRPV4 contributes to sunburn pain [11] and epidermal channel expression is deregulated in some conditions that cause pathological pain [12].

Progress in characterizing skin-nociceptor interactions has been limited by the heterogeneity and complexity of mammalian systems. Although scRNA-seq studies are rapidly expanding the molecular taxonomy of SSNs [13–15], measures of mammalian SSN diversity remain understudied. Likewise, mammalian skin varies in cellular composition across anatomical locations [16–18], as do innervation patterns of SSNs [19,20]. We therefore set out to characterize skin-nociceptor interactions in a more tractable experimental system, *Drosophila* larvae. In *Drosophila*, a single class of identified SSNs, Class IV dendrite arborization (C4da) neurons, are necessary and sufficient for nociception: inactivating C4da neurons renders flies insensitive to noxious stimuli and activating these neurons drives nociceptive behavior responses [21]. Dendrites of C4da neurons densely innervate the larval body wall, growing along the basal surface of an epidermis comprised primarily of three cell types: a monolayer of ∼1000 tiled epidermal cells per hemi-segment interspersed with apodemes, specialized epidermal cells that serve as sites of body wall muscle attachment, and histoblasts, stem cells that repopulate the epidermis after metamorphosis [4].

Here we report the identification of the microRNA *miR-14* as a factor that provides both selectivity and specificity to dendrite-epidermis interactions. Dendrites of nociceptive C4da neurons differentially arborize over apodeme and non-apodeme epidermal cells, intercalating between apodemes but not other epidermal cells. This differential pattern of arborization is controlled in part by *miR-14*, which is expressed throughout the epidermis with the notable exception of apodemes. *miR-14* functions in epidermal cells to restrict dendrite intercalation by C4da neurons, but not other SSNs, doing so through control of gap junctions. Finally, we find that mechanically evoked nociceptive responses are tuned according to the extent of epidermal dendrite intercalation: the increased intercalation in *miR-14* mutants drives enhanced nocifensive responses to mechanical stimuli and heightened mechanosensory responses in C4da neurons, while eliminating dendrite intercalation at apodemes has the opposite effect.

## Results

### The miRNA *miR-14* regulates dendrite orientation over epidermal cells

C4da dendrites adopt distinct innervation patterns in territory populated by apodemes versus other epidermal cells: dendrites spread extensively over the basal surface of most epidermal cells without aligning to their cell-cell interfaces, but align along apodeme cell-cell interfaces and avoid innervating territory beneath apodemes (Fig 1A–1D). These distinct orientations likely reflect physical occlusion of apodeme territory by muscles which attach to their tendon cells during embryonic development several hours prior to the sprouting of sensory neuron dendrites [22,23]. However, experimental evidence supporting this idea is lacking. Furthermore, whether there are attractive cues at apodeme junctional domains, and/or the repulsive cues at other epidermal cell-cell interfaces is currently unknown. To identify spatial cues that direct these distinct C4da dendrite innervation patterns, we screened EMS-induced larval lethal alleles [24] for mutations that differentially affected dendrite orientation over apodemes and other epidermal cells. We identified a single mutant allele (*dendrite growth 29*, *dg29*) in the gene *Dicer1* (*Dcr1*) that caused two interrelated defects in dendrite patterning. First, *Dcr1$^{dg29}$* mutants exhibited a significant increase in dendrite alignment to epidermal intercellular junctions outside of apodeme domains without affecting dendrite orientation over apodemes (S1A–S1E Fig). Second, dendrites in *Dcr1$^{dg29}$* mutants exhibited a significant increase in dendrite-dendrite crossing (S1F Fig), which frequently occurred at sites of junctional dendrite alignment in both control and *Dcr1$^{dg29}$* mutant larvae (S1G and S1H Fig).

*Dcr1* encodes an enzyme required for pre-miRNA processing [25,26], and prior studies defined roles for miRNAs in C4da dendrite scaling growth and terminal dendrite growth [27,28]. We therefore hypothesized that *Dcr1$^{dg29}$* dendrite patterning defects reflected requirements for multiple miRNAs including one or more that controlled epidermal junctional alignment. Indeed, a comprehensive screen of miRNA deficiency alleles (S1I Fig) revealed that mutation in a single miRNA gene, *miR-14*, caused dendrite alignment defects comparable to *Dcr$^{dg29}$* (S1J–S1P Fig).

To directly visualize dendrite-epidermis interactions in *miR-14* mutants we labeled C4da dendrites with *ppk-CD4-tdTomato* and epidermal membranes with the phosphatidylinositol 4,5-bisphosphate (PIP$_2$) reporter PLCδ-PH-GFP that accumulates at epidermal cell-cell junctions and additionally labels sites of epidermal dendrite ensheathment [5,29]. Compared to wild type controls, *miR-14* mutants exhibited several unique features with respect to dendrite positioning. First, *miR-14* mutants had an increased incidence of junctional dendrite alignment, with 29% of *miR-14* mutant C4da dendrites aligned along epidermal junctions compared to 4% in wild-type controls (Fig 1E–1G). As with *Dcr$^{dg29}$* mutants, *miR-14* mutation did not affect dendrite orientation over apodemes (S2A Fig). Second, aligned dendrites in *miR-14*

*Luedke et al,*

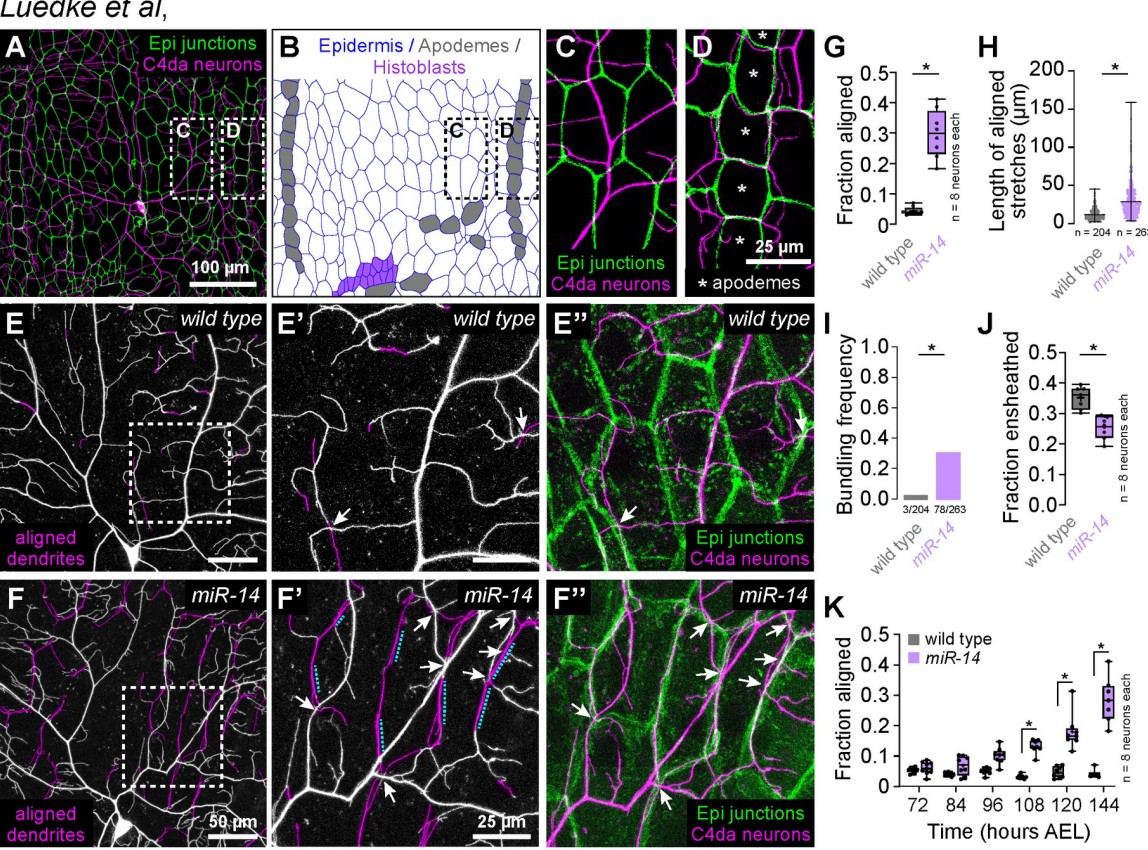

**Fig 1. Identification of a genetic program that limits dendrite alignment along epidermal cell-cell junctions.** (A-D) Dual-labeling of epidermal septate junctions (*Nrx-IV^GFP*) and nociceptive C4da neurons (*ppk-CD4-tdTomato*) in third instar larvae. (A) Maximum projection of a confocal stack showing the distribution of C4da dendrites over epidermal cells in a single dorsal hemisegment. (B) Tracing of epidermal cells depicting distribution of the principal epidermal cell types (epidermal cells, apodemes and histoblasts, pseudocolored as indicated). (C-D) High magnification views of dendrite position over epidermal cells (C) adjacent to posterior segment boundary and (D) dendrite position over apodemes (marked with asterisks) at the posterior segment boundary. Asterisks mark apodemes. (E-I) *miR-14* regulates dendrite position over epidermal cells. (E-F) Maximum intensity projections of C4da neurons from (E) wild-type control and (F) *miR-14^Δ1* mutant larvae at 120 h AEL showing epidermal junction-aligned dendrites pseudocolored in magenta. Insets (E' and F') show high magnification views of corresponding regions from control and *miR-14^Δ1* mutants containing junction-aligned dendrites. Arrows mark dendrite crossing events involving junction-aligned dendrites and dashed lines mark aligned dendrites that are bundled. (E" and F") Maximum intensity projections show relative positions of C4da neurons (*ppk-CD4-tdTomato*) and epidermal junctions labeled by the PIP2 marker PLC^δ-PH-GFP. (G-K) Quantification of epidermal dendrite alignment phenotypes. Plots depict (G) the fraction of C4da dendrite arbors that are aligned along epidermal junctions at 120 h AEL, (H) the length of aligned stretches of C4da dendrites, (I) the proportion of terminal dendrites that were present in bundles, (J) the proportion of C4da dendrites ensheathed by epidermal cells at 120 h AEL, and (I) the proportion of C4da dendrite arbors that aligned along epidermal junctions over a developmental time course in control and *miR-14^Δ1* mutant larvae. Box plots here and in subsequent panels depict mean values and 1st/ 3rd quartile, whiskers mark 1.5x IQR, and individual data points are shown. *P<0.05 compared to pre-EMS control; Mann Whitney test (G-H), Fisher's exact test (I), unpaired t-test with Welch's correction (J) or Kruskal-Wallis test with post-hoc BH correction (K).. Sample sizes are indicated in each panel.

mutants tracked epidermal junctions over extended length scales, often spanning multiple epidermal cells, whereas control dendrites aligned to junctions only over short stretches (Fig 1E, 1F and 1H). Dendrite spread over the epidermis was therefore limited outside of junctional domains in *miR-14* mutants, resulting in a significant reduction in overall body wall coverage by C4da dendrites (S1Q Fig). Third, multiple branches frequently bundled together at sites of epidermal junction alignment in *miR-14* mutants; this bundling was rarely observed in controls (Fig 1F and 1I). Finally, *miR-14* mutants first exhibited elevated levels of junctional

dendrite alignment at 108 h AEL (Fig 1K), more than two days after C4da dendrites establish complete coverage of the body wall [27], suggesting that the aberrant alignment results not from junctional targeting during primary dendrite outgrowth but from arbor repositioning during later larval development.

Dendrite arbors of C4da neurons become progressively ensheathed by epidermal cells during larval development [30,31], and the extent of epidermal junction alignment by *miR-14* mutant C4da dendrites was comparable to the extent of epidermal dendrite ensheathment in wild-type larvae [5]. We therefore examined the relationship between these two epidermis-SSN interactions. *miR-14* mutants exhibited a modest reduction in C4da dendrite ensheathment outside of junctional domains (Fig 1J), but several observations suggest that junctional alignment and epidermal ensheathment of dendrites are distinct phenomena. First, these two epidermis-SSN interactions map to different portions of the dendrite arbor: junctional alignment primarily involves terminal dendrites (S2C and S2D Fig), which are rarely ensheathed [5]. Second, different types of da neurons are ensheathed to different degrees, but junctional dendrite alignment selectively occurs in C4da neurons: we observed negligible alignment of dendrite from C1da or C3da neurons to epidermal junctions in either wild-type control or *miR-14* mutant larvae (S2E–S2H Fig). Third, ensheathment and junctional alignment are genetically separable: mutation in the microRNA *bantam* (*ban*) blocks epidermal dendrite ensheathment [32], but not junctional dendrite alignment (S2I and S2J Fig). Fourth, these two dendrite-epidermis interactions occur at different developmental times: ensheathment progressively increases after the 1st/2nd instar larval transition [5], whereas junctional alignment occurs at constant level in control larvae but inappropriately increases in *miR-14* 3rd instar larvae (S2B Fig). Finally, epidermal sheaths form on the basal surface of individual epidermal cells and terminate at junctional domains [5] whereas junction-aligned dendrites tracked multiple epidermal cells in *miR-14* mutants (Fig 1). Hence, we conclude that junctional alignment and ensheathment are two distinct types of epidermis-dendrite interactions.

## *miR-14* antagonizes formation and elongation of junction-aligned dendrites

We used time-lapse imaging to identify the developmental origin of dendrite alignment defects in *miR-14* mutants, focusing on growth dynamics of existing junction-aligned dendrites, rates of addition and loss of junctional dendrite alignment, and the orientation of new dendrite outgrowth with respect to epidermal junctions. Epidermal junction-aligned dendrites exhibited equivalent rates of growth and retraction in control larvae (Fig 2A and 2C), consistent with the observation that the proportion of dendrite arbors aligned to epidermal junctions is unchanged during this developmental window (Figs 1K and S2B). In contrast, junction-aligned dendrites exhibited significantly more growth than retraction and the magnitude of growth events but not retraction events was significantly larger in *miR-14* mutants than in controls (Fig 2B–2D). *miR-14* mutant C4da neurons likewise exhibited an increased incidence of new junctional dendrite alignment events during the time lapse (Fig 2E). Finally, newly aligned dendrites originated from new dendrite growth events rather than displacement of existing dendrites in both wild-type controls and *miR-14* mutants (Fig 2F).

The increased frequency of new branch alignment in *miR-14* mutants could reflect oriented growth towards epidermal junctions, increased accessibility of junctions to dendrites, or both. To distinguish between these possibilities, we monitored the orientation of newly formed dendrite branches with respect to epidermal cell-cell junctions (S3 Fig). Compared to controls, a significantly larger portion of new branches that originated at epidermal cell-cell interfaces oriented along junctions in *miR-14* mutants (Fig 2G). In contrast, branches that originated

*Luedke et al,*

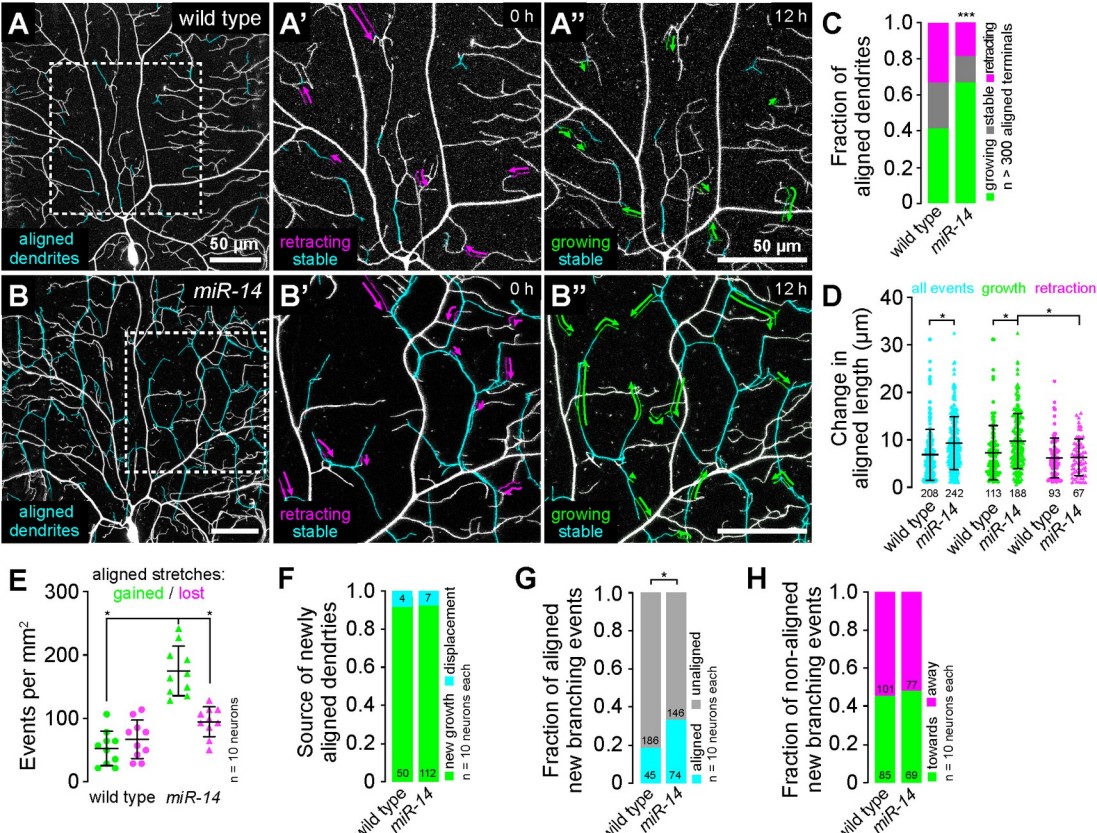

**Fig 2. Time-lapse analysis of dendrite-epidermal junction interactions.** C4da neurons (*ppk-CD4-tdTomato*) were imaged over a 12 h time-lapse (108–120 h AEL) and dynamics were monitored for junction-aligned dendrites identified via co-localization with an epidermal junction marker (*A58-GAL4, UAS-PLC$^\delta$-PH-GFP*). Epidermal junction-aligned dendrites were pseudocolored cyan in the initial time point, and growth (green) and retraction (magenta) were pseudocolored in a composite of the two time points. Representative images are shown for (A) wild-type control and (B) *miR-14* mutant larvae. (C) Stacked bar plot shows the fraction of junction-aligned dendrites that were growing, stable, or retracting over the time-lapse. (D) Extent of dynamics. Plot shows the change in length for each aligned dendrite measured (points) as well as mean and standard deviation. (E) Frequency of turnover in junctional alignment. Bars depict mean and standard deviation and data points represent the number of junctional-alignment events gained (green) or lost (magenta) during the time lapse for individual neurons, normalized to the area sampled. (F-H) Time-lapse imaging of new dendrite branch alignment relative to epidermal junctions. C4da neurons were imaged over a 24 h time lapse (96–120 h AEL) and the orientation of dendrite branch growth relative to epidermal junctions was monitored for each new dendrite branch. (F) Bars depict the proportion of newly aligned dendrite stretches (aligned to epidermal junctions at 120 h but not 96 h) that involve new dendrite growth (green) or reorientation of existing dendrites (cyan). Chi-square analysis revealed no significant difference between wild-type controls and *miR-14* mutants. (G) A significantly larger proportion of new dendrite branches (present at 120 h but not 96 h) align along epidermal junctions in *miR-14* mutants compared to wild-type controls. (H) Comparable portions of unaligned new dendrite branches in *miR-14* mutants and wild-type controls orient towards (green) and away from (magenta) the nearest epidermal cell-cell interface. *P<0.05 compared to wild-type control unless otherwise indicated, Chi-square analysis (C, F-H), Kruskal-Wallis test with post-hoc Dunn's test (D), or one-way ANOVA with post-hoc Sidak's test (E).

outside of junctional domains grew towards and away from epidermal junctions with equivalent frequencies in *miR-14* mutants and wild-type controls (Fig 2H). These results are consistent with a model in which increased accessibility to epidermal intercellular space and/or junction-localized adhesive cues drive junctional alignment in *miR-14* mutants.

## Junction-aligned dendrites apically intercalate between epidermal cells

Epidermal junction-aligned dendrites frequently engage in dendrite crossing, which in other contexts involves apical dendrite detachment from the extracellular matrix (ECM) [30,31]. We

therefore investigated whether junction-aligned dendrites apically intercalate between epidermal cells to facilitate out-of-plane crossing of unaligned dendrites (Fig 3A). First, we measured apical dendrite displacement from the ECM by monitoring co-localization of C4da dendrites (*ppk-CD4-tdTomato*) with the epidermal basement membrane (BM) marker *trol-GFP* [33]. *miR-14* mutant dendrites exhibited alterations in the frequency and distribution of dendrite detachment: nearly 30% of the *miR-14* mutant C4da arbor was apically displaced from the BM, compared with ∼8% in control larvae (Fig 3B and 3C) and this apical displacement primarily involved terminal dendrites, which are also the primary source of junction-aligned dendrites (Figs 3D and S2D). Next, we monitored C4da dendrite axial position relative to epidermal cells expressing cytosolic red fluorescent protein (RFP) to label the entire epidermal cell volume. In wild-type larvae, dendrites rarely aligned to epidermal junctions and were restricted to the basal epidermal surface at junctional domains (Fig 3E). In contrast, *miR-14* mutant dendrites frequently aligned to epidermal junctions and penetrated to the apical surface (Fig 3F).

We corroborated these results with high-resolution confocal imaging of C4da dendrites in larvae carrying an mCherry tagged allele of *shotgun* (*shg^mCherry*) to visualize epidermal adherens junctions (AJs) and found that epidermal junction-aligned C4da dendrites targeted apical domains and engaged in dendrite-dendrite crossing with basally localized dendrites (Fig 3G and 3H). Furthermore, in control and *miR-14* mutant larvae we observed an inverse correlation between dendrite axial position and the angle at which dendrites encountered epidermal junctions: dendrites crossing epidermal junctions at normal angles were positioned the furthest from AJs, typically 3 μm or more, and those crossing at glancing angles or aligned to junctions were typically positioned within 1.5 μm of AJs (Fig 3I). Hence, junction-aligned dendrites apically intercalate between epidermal cells.

To determine whether dendrite intercalation exhibited positional bias in the epidermis, we monitored two features of epidermal cells containing junction-aligned dendrites. First, we examined whether the probability of dendrite intercalation co-varied with length of epidermal cell-cell interfaces and hence epidermal cell size, but we observed no bias for dendrite intercalation to a particular epidermal edge length (Fig 3J). Second, we assayed for bias in the lateral position of dendrite insertion into epidermal junctions. We found that epidermal dendrite intercalation most frequently occurred in proximity to tricellular junctions, the point at which three epidermal cells contact (Fig 3K), possibly reflecting an increase in accessibility and/or enrichment of factors that promote apical dendrite targeting at these sites in *miR-14* mutants.

## *miR-14* functions in epidermal cells to limit junctional dendrite intercalation

To define the site of action for *miR-14* control of dendrite-epidermis interactions we examined whether *miR-14* is expressed in C4da neurons, epidermal cells, or both. First, we generated a transcriptional reporter (*miR-14-GAL4*) containing ∼3 kb of the *miR-14* promoter driving expression of GAL4. Using *miR-14-GAL4* to drive *miR-14* expression rescued the junctional dendrite alignment defect of *miR-14* mutants (S4A–S4C Fig), demonstrating that this reporter encompasses the *miR-14* expression domain required for larval sensory dendrite positioning. Next, we monitored reporter expression within the larval body wall and found that *miR-14-GAL4* was prominently expressed in epidermal cells and stochastically expressed in a subset of SSNs that did not include C4da neurons (Fig 4A). Within the epidermis, *miR-14-GAL4* expression was largely absent from apodemes (Fig 4A and 4B), which are the primary sites of junctional dendrite alignment in wild-type larvae.

To extend our expression studies, we used a miRNA sensor to monitor the spatial distribution of *miR-14* activity [34]. The sensor transgene consists of a ubiquitous promoter driving

*Luedke et al,*

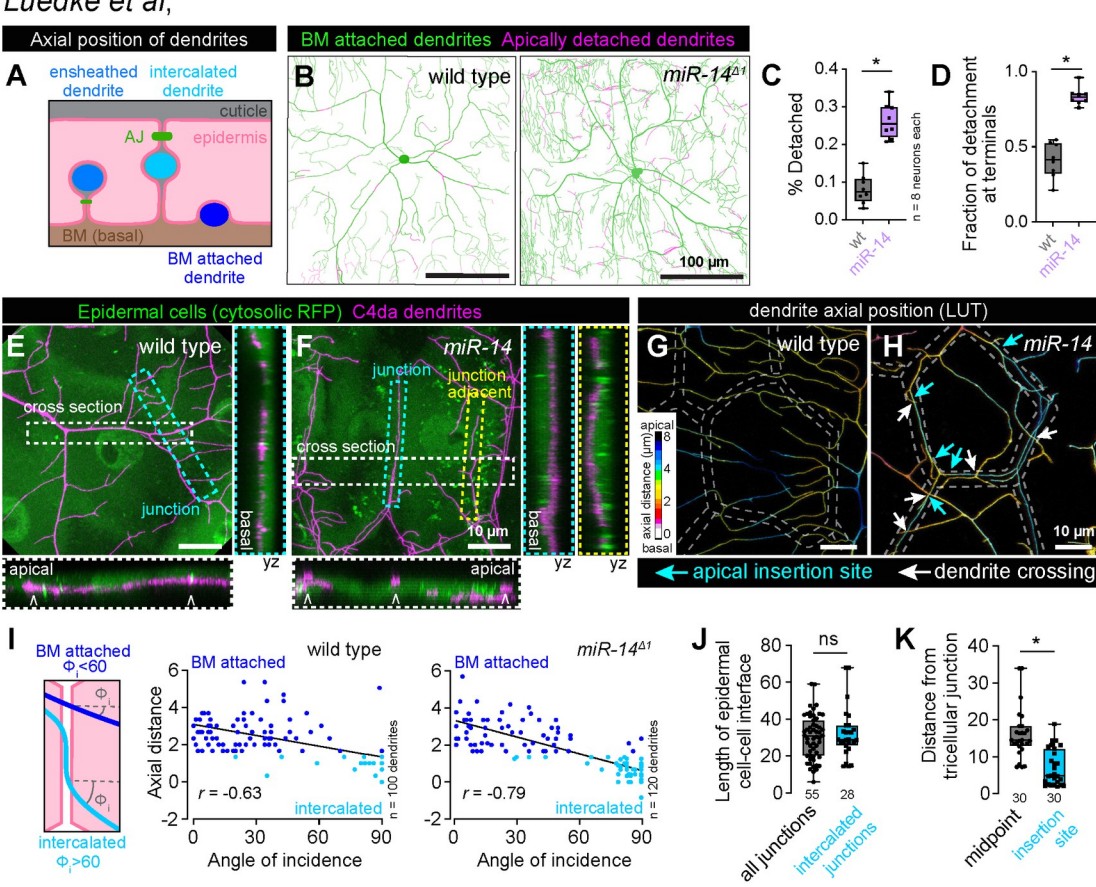

**Fig 3. Junctional-aligned dendrites intercalate between epidermal cells.** (A) Schematic illustrating the axial position of ensheathed, epidermal junction aligned (intercalated) and BM-attached dendrites. (B) Apical dendrite detachment from the BM. Traces depict BM-contacting dendrites in green and BM-detached dendrites in magenta for representative wild-type control and *miR-14* mutant C4da neurons. Plots depict (C) the fraction of C4da dendrites apically detached from the BM and (D) the fraction of apical detachment that involves terminal dendrites. *P<0.05 compared to wild-type control, unpaired t-test with Welch's correction. (E-H) Junction-aligned dendrites apically intercalate between epidermal cells. Maximum intensity projections show distribution of C4da dendrites over individual epidermal cells in wild-type control (E) and *miR-14* mutant larvae (F). Orthogonal sections span the width of epidermal cells (cross section; carets mark position of epidermal junctions), the epidermal cell-cell interface (junction section, marked by cyan hatched box) and run adjacent to the epidermal cell-cell interface (junction adjacent section, marked by yellow hatched box). Among these, only junction-aligned dendrites penetrate to the apical surface of epidermal cells. (G-H) Z-projections of confocal stacks depicting axial dendrite position according to a lookup table in wild-type control (G) and *miR-14* mutant larvae (H). White hatched lines outline epidermal junctions, white arrows depict dendrite crossing events involving junction-aligned dendrites, cyan arrows depict apical insertion site of junction-aligned dendrites (I) Schematic depicting approach to measuring dendrite-epidermal junction angles of incidence (left) and scatterplots of axial distance (dendrite to epidermal AJ) versus dendrite-junction angle of incidence (right). Note the inverse linear regression (black lines). (J) Dendrite intercalation is distributed across a range of epidermal cell sizes but preferentially occurs near tricellular junctions. The plot depicts the distribution of edge lengths at epidermal cell-cell interfaces (all junctions, gray) and the length distribution of all epidermal cell-cell interfaces that contain intercalated dendrites (intercalated junctions, cyan). (K) The insertion site for epidermal intercalation is biased towards tricellular junctions. The plot depicts the length from the midpoint of the cell-cell interface to a tri-cellular junction (midpoint) and the distance between the site of apical dendrite intercalation from the nearest tricellular epidermal junction. Measurements were taken from 30 epidermal cell-cell interfaces in *miR-14* mutant larvae that contained a single intercalated dendrite. *P<0.05, ns, not significant (P>0.05) compared to wild-type control, Wilcoxon rank sum test.

GFP expression and a 3' UTR containing *miR-14* binding sites, hence GFP expression is attenuated in cells where the miRNA is active. A control sensor lacking *miR-14* binding sites was expressed throughout the epidermis, with apodemes and other epidermal cells exhibiting comparable levels of GFP fluorescence (Fig 4C and 4E). In contrast, *miR-14* sensor expression was

**Fig 4. *miR-14* functions in epidermal cells to control dendrite position.** (A-B) *miR-14-GAL4* is highly expressed in the epidermis with the exception of apodemes, where expression is largely absent. (A) Maximum intensity projection shows *miR-14-GAL4, UAS-tdTomato* expression in the body wall of third instar larvae additionally expressing *Nrg$^{GFP}$* to label epidermal junctions. Dashed lines mark apodemes, asterisks mark cells lacking detectable *miR-14-GAL4* expression and arrowheads mark lowly expressing cells. (B) Stacked bars depict the proportion of epidermal cells (n = 725 cells) and apodemes (n = 149 cells) with the indicated levels of *miR-14-GAL4* expression. *P<0.05, Chi-square test. (C-E) *miR-14* activity mirrors *miR-14-GAL4* expression patterns. Maximum intensity projections depicting (C) control or (D) *miR-14* activity sensor in larvae additionally expressing *sr-GAL4, UAS-mCD8-RFP* to label apodemes are shown (left) along with lookup tables depicting miRNA sensor intensity (right). Insets show miRNA sensor design and dashed lines outline apodemes. (E) *miR-14* sensor expression, which is inversely related to *miR-14* activity, is attenuated throughout the epidermis with the exception of apodemes. NS, not significant, *P<0.05, Kruskal-Wallis test followed by Dunn's multiple comparisons test. (F) *miR-14* is dispensable in C4da neurons for dendrite morphogenesis. Total dendrite length and the number of dendrite crossings in *miR-14* or *drosha* mutant C4da MARCM clones are indistinguishable from wild-type controls. NS, not significant, Kruskal-Wallis test followed by Dunn's multiple comparisons test. (G) Epidermal *miR-14* function is required for proper dendrite positioning. Dendrite crossing frequency is shown for larvae expressing control or *miR-14* sponge transgenes using the indicated GAL4 drivers. *P<0.05, ANOVA with post-hoc Sidak's test. (H) Epidermal *miR-14* expression is sufficient to support proper dendrite positioning. Dendrite crossing frequency is shown for *miR-14* mutant larvae expressing *UAS-miR-14* under control of the indicated GAL4 drivers. NS, not significant, *P<0.05, ANOVA with post-hoc Sidak's test.

largely attenuated in the epidermis, with the notable exception of apodemes, consistent with our observation that *miR-14* expression is limited in apodemes (Fig 4D and 4E). Taken together, our expression studies support a model in which *miR-14* is active in epidermal cells but not apodemes to restrict epidermal dendrite intercalation.

To directly test tissue-specific requirements for *miR-14* we used mosaic genetic analysis and monitored dendrite crossing as a proxy for epidermal dendrite intercalation (S1F–S1H Fig). First, we generated single C4da neuron clones homozygous for a *miR-14* null mutation in a

heterozygous background using MARCM [35]. We found that *miR-14* mutant and wild-type control C4da neuron MARCM clones exhibited comparable dendrite branch number, overall dendrite length, and dendrite crossing events, suggesting that *miR-14* is dispensable in C4da neurons for dendrite morphogenesis (Figs 4F and S4D–S4E). Similarly, C4da neuron MARCM clones carrying a null mutation in *drosha*, which encodes a ribonuclease required for miRNA processing [36], exhibited no significant increase in dendrite-dendrite crossing (Figs 4F and S4F), consistent with the model that *miR-14* functions neuron non-autonomously to control C4da dendrite position.

To assay epidermal requirements for *miR-14* we used a miRNA sponge to reduce the bio-available pool of *miR-14*. This miRNA sponge carries synthetic *miR-14* binding sites with perfect complementarity in the miRNA seed region and bulges at the Argonaute cleavage site, hence the binding sites stably interact with *miR-14* and act as competitive inhibitors [37,38]. Ubiquitous (*Actin-GAL4*) expression of the *miR-14* sponge but not a control sponge with scrambled *miR-14* binding sites phenocopied dendrite defects of *miR-14* null mutants, demonstrating the efficacy of the approach (Figs 4G, S4G and S4J). Selective sponge expression in epidermal cells (*A58-GAL4*) but not sensory neurons (*5-40-GAL4*) yielded similar results (Figs 4G, S4H, S4I, S4K and S4L), demonstrating that *miR-14* is necessary in epidermal cells but dispensable in sensory neurons for control of dendrite position.

Next, we used genetic rescue assays to determine whether selective *miR-14* expression in epidermal cells or sensory neurons suppressed *miR-14* mutant dendrite positioning defects. For these assays we utilized a *UAS-Luciferase* transgene carrying functional *miR-14* stem loops in the 3' UTR that are cleaved following transcription [39]. Ubiquitous (*Actin-GAL4*) or epidermal (*A58-GAL4*) but not neuronal (*ppk-GAL4*) expression of *UAS-Luciferase-miR-14* rescued the *miR-14* mutant dendrite crossing defects (Figs 4H and S4M–S4P), consistent with the model that *miR-14* acts in epidermal cells to control dendrite position. We note that rescue of *miR-14* dendrite crossing defects was incomplete with epidermal *UAS-Luciferase-miR-14* expression; this could reflect requirements for *miR-14* in other tissues or differences in the expression levels/timing within the epidermis provided by the *Actin-GAL4* and *A58-GAL4* drivers. We favor the latter as we have not uncovered requirements for *miR-14* in other peripheral cell types, or additive effects when altering *miR-14* expression simultaneously in epidermal cells and C4da neurons.

Our studies thus far indicate that *miR-14* acts in epidermal cells to prevent dendrites from intercalating between epidermal cells, an orientation that resembles dendrite orientation over apodemes. We therefore hypothesized that *miR-14* functions in epidermal cells to restrict an apodeme-like cell fate. To explore this possibility, we used RNA-seq analysis to assay for *miR-14* control of apodeme marker gene expression in epidermal cells. First, to define a panel of apodeme marker genes, we dissociated fillet preparations of larvae bearing fluorescent markers for apodemes (*sr-GAL4, UAS-mCD4-tdGFP / +*) or epidermal cells (*R38F11-GAL4, UAS-mCD4-tdGFP / +*) into single cell suspensions and subjected 10-cell pools of manually picked apodemes and epidermal cells to RNA-seq analysis. We identified 383 transcripts that were differentially expressed in apodemes and other epidermal cells (S5A and S5B Fig and S1 Table), including the previously identified apodeme markers *beta-Tubulin56D*, *held out wings*, *Leucine-rich tendon-specific protein*, *stripe*, *Tiggrin*, and *Thrombospondin* [40]. Next, we used RNA-seq to assay for gene expression changes in *miR-14* mutant epidermal cells, focusing on the panel of differentially expressed apodeme genes (apodeme DEGs). Although we identified 65 transcripts that were differentially expressed in *miR-14* mutant epidermal cells when compared to controls (S5B Fig and S1 Table), *miR-14* had no effect on expression of known apodeme marker genes and affected expression of only 5/383 apodeme DEGs overall. Hence, *miR-14* does not appear to function as a molecular switch to repress apodeme cell fate.

Next, we investigated whether ectopic *miR-14* expression in apodemes prevented epidermal intercalation. For these experiments we used the *sr-GAL4* driver for selective apodeme expression of *UAS-DsRed* or *UAS-DsRed-miR-14*, a *UAS-DsRed* transgene bearing multiple copies of *miR-14* in the 3'UTR and monitored C4da dendrite distribution over the RFP-labeled apodemes. In control larvae, C4da dendrites were largely excluded from the basal surface of apodemes: they accumulated along apodeme cell boundaries, coalescing at apodeme-apodeme junctions where they intercalated, and often wrapped around apodemes upon exiting the intercellular domain (S5C Fig). In contrast, ectopic expression of *miR-14* in apodemes induced a significant increase in dendrite incursion into apodeme domains (S5D and S5E Fig). Taken together, these studies are consistent with a model in which *miR-14* regulates intercellular interactions that promote dendrite intercalation.

## *miR-14* regulates mechanical nociceptive sensitivity

Epidermal ensheathment modulates sensitivity to noxious mechanosensory inputs [5], therefore we hypothesized that epidermal dendrite intercalation would likewise influence nociceptor sensitivity to mechanical stimuli. Harsh touch activates C4da neurons to elicit nocifensive rolling responses [41], and indeed we found that a single 80 millinewton (mN) poke elicited nocifensive rolling responses in 72% of control larvae (Figs 5A and S6A). In contrast, *miR-14* mutant larvae exhibited an increased frequency of nocifensive responses to the same stimulus (88% roll probability, Fig 5A), and this enhancement was more pronounced with reduced forces. For example, 25 mN stimulation yielded nociceptive responses in 20% of control and 63% of *miR-14* mutant larvae, and this was accompanied by an increased magnitude of response (Fig 5A and 5B). We observed comparable effects on mechanical nociception with *Dcr*$^{dg29}$ and an additional allele of *miR-14* (S6B Fig) and found that *UAS-Luciferase-miR-14* expression with *miR-14-GAL4* rescued the mechanical sensitization in *miR-14* mutants (S6C Fig), further demonstrating that loss of *miR-14* induces mechanical nociceptive sensitization.

*miR-14* functions in epidermal cells to control dendrite position of nociceptive C4da neurons but not other larval SSNs, therefore we examined whether *miR-14* exhibited the same selectivity in control of mechanosensory behavior. First, we assayed *miR-14* mutant responses to non-noxious mechanosensory stimuli, including gentle touch responses mediated by C3da neurons [42] and vibration responses mediated by chordotonal neurons [43]. Unlike noxious touch, gentle touch and vibration stimuli elicited responses in *miR-14* mutants that were indistinguishable from wild-type controls (Fig 5C and 5D). Second, we investigated whether *miR-14* functions in epidermal cells to control mechanical nociceptive sensitivity. Indeed, selective epidermal *miR-14* inactivation using a *miR-14* sponge induced mechanical hypersensitivity, whereas resupplying *miR-14* expression selectively to epidermal cells suppressed the *miR-14* mutant mechanical hypersensitivity phenotype (Fig 5F and 5G). Hence, *miR-14* functions in epidermal cells where it selectively influences C4da neuron form and function.

C4da neurons are polymodal nociceptors that respond to mechanical, thermal, chemical, and ultraviolet stimuli. We next investigated whether *miR-14* mutation broadly sensitized larvae to noxious stimuli. For these studies we focused on behavioral responses to noxious heat [44] and assayed whether *miR-14* mutation altered the nociceptive rolling response probability or latency. We found that control and *miR-14* mutant larvae exhibited comparable thermal nociceptive responses over a range of noxious temperatures (Fig 5E). Hence, *miR-14* mutation sensitizes larvae to noxious mechanical but not noxious heat stimuli.

Tissue damage and chemical toxins sensitize *Drosophila* larvae to noxious stimuli, so we next investigated the relationship between *miR-14* and known mechanisms of nociceptive sensitization. First, UV damage induces thermal allodynia and hyperalgesia in *Drosophila* that is

*Luedke et al,*

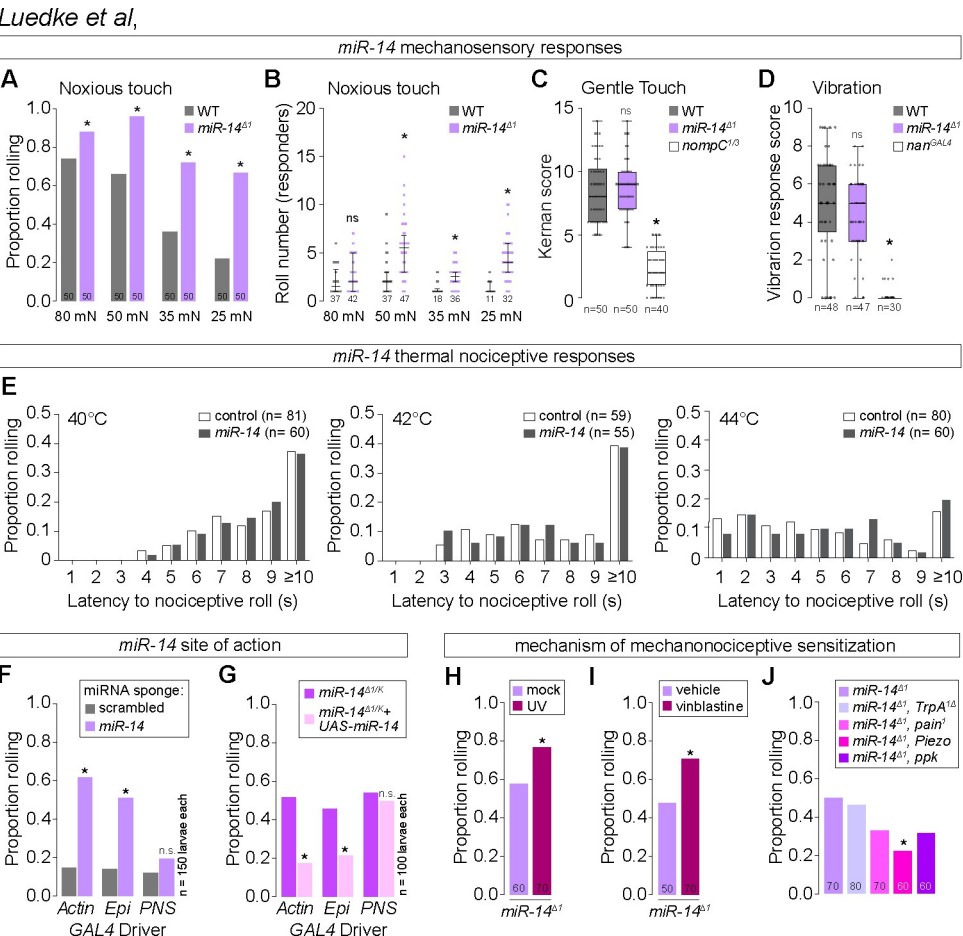

**Fig 5. *miR-14* regulates sensitivity to noxious mechanical inputs.** (A-B) *miR-14* mutants exhibit enhanced nocifensive behavior responses. Plots depict (A) proportion of larvae that exhibit nocifensive rolling responses to von Frey fiber stimulation of the indicated intensities and (B) mean number of nocifensive rolls. (C-D) *miR-14* mutation does not affect larval responses to non-noxious mechanical stimuli. Plots depict larval responses to (C) gentle touch and (D) vibration for larvae of the indicated genotypes. (E) Plots depict nocifensive rolling response probability (y-axis) and latency (x-axis) of control and *miR-14* mutant larvae to the indicated thermal stimuli. (F-G) *miR-14* functions in epidermal cells to control mechanical nociceptive sensitivity. (F) Plot depicts nocifensive rolling responses to 25 mN von Frey fiber stimulation of larvae expressing a *miR-14* sponge or a control sponge with scrambled *miR-14* binding sites under control of the indicated *GAL4* driver (Ubiq, ubiquitous expression via *Actin-GAL4*; Epi, epidermal expression via *A58-GAL4*; PNS, md neuron expression via *5-40-GAL4*). (G) Plot depicts nocifensive rolling responses to 25 mN von Frey fiber stimulation of *miR-14* mutant larvae expressing the indicated *GAL4* drivers with or without *UAS-miR-14*. (H-J) *miR-14* acts independent of known pathways for nociceptive sensitization. Plots depict nocifensive rolling responses to 25 mN von Frey fiber stimulation of control or *miR-14* mutant larvae (G) 24 h following mock treatment or UV irradiation and (H) following 24 h of feeding vehicle or vinblastine, or (I) of *miR-14* mutant larvae carrying loss-of-function mutations in the indicated sensory channels. NS, not significant, *P<0.05, Fisher's exact test with a BH correction (A, H-J), Kruskal-Wallis test followed by a Dunn's multiple comparisons test, or Chi-square test (G). (B-D). The number of larvae tested is shown for each condition.

triggered in part by epidermal release of the inflammatory cytokine eiger [7]. We found that *miR-14* mutation and UV damage had additive effects on mechanical nociceptive sensitivity (Fig 5H), whereas *miR-14* mutation had no effect on UV-induced thermal hyperalgesia (S6D Fig). Furthermore, epidermal knockdown of *eiger*, which attenuates UV-induced nociceptive sensitization [7], had no effect on *miR-14* mutant mechanonociceptive responses (S6E Fig). Second, the chemotherapeutic agent vinblastine induces mechanical allodynia in both invertebrates and vertebrates [45], and we found that *miR-14* mutation and vinblastine feeding had

additive effects on mechanical nociceptive responses (Fig 5I). Finally, whereas nociceptive sensitization induced by UV damage and vinblastine requires *TrpA1* [45,46], we found that *TrpA1* is dispensable for *miR-14*-dependent sensitization (Fig 5J). Instead, mutations in other mechanosensory channel genes, principally *Piezo*, suppressed *miR-14* mutant mechanonociception defects without affecting dendrite intercalation (Figs 5J and S6F–S6I), consistent with a model in which epidermal intercalation potentiates mechanically evoked nocifensive responses via Piezo. Taken together, our results demonstrate that *miR-14* induces mechanical hypersensitivity independent of known mechanisms of nociceptive sensitization.

## Epidermal dendrite intercalation promotes nociceptive sensitization

To probe the relationship between epidermal dendrite intercalation and mechanical nociceptive sensitivity, we examined whether suppressing dendrite intercalation affected *miR-14* mutant responses to noxious mechanical stimuli. Ensheathed portions of C4da dendrite arbors are apically displaced inside epidermal cells, however formation of these sheaths and the accompanying apical dendrite displacement are suppressed by neuronal overexpression of integrins [31]. We hypothesized that integrin overexpression would likewise suppress apical dendrite intercalation between epidermal cells, and indeed we found that co-overexpression of alpha (*UAS-mew*) and beta (*UAS-mys*) integrin subunits in C4da neurons significantly reduced junctional dendrite alignment in *miR-14* mutants without altering other aspects of dendrite morphogenesis (Fig 6A–6F). Neuronal integrin overexpression had corresponding effects on nociceptive sensitization: responses to noxious mechanical stimuli were significantly attenuated by overexpressing integrins in *miR-14* mutants but not in controls (Fig 6G). We note that neuronal integrin overexpression in control larvae attenuated responses to 80 mN stimuli (S7A Fig), consistent with prior results demonstrating that blocking epidermal ensheathment attenuates responses to high intensity noxious stimuli [5]. However, integrin overexpression had no impact on responses of control larvae to 20 mN stimuli, suggesting that epidermal dendrite intercalation and epidermal ensheathment differentially impact larval sensitivity to noxious mechanical cues.

To further examine the relationship between epidermal dendrite intercalation and mechanical nociceptive sensitivity we conducted a genetic modifier screen for EMS-induced mutations that altered the extent of epidermal dendrite intercalation in *miR-14* mutants. From this screen we identified one enhancer mutation (*mda-1*, *modifier of miR-14 dendrite alignment*) that significantly increased the extent of epidermal dendrite alignment in *miR-14* mutants (S7A–S7C Fig) as well as nociceptive sensitivity (S7D Fig), further underscoring the correlation between epidermal dendrite intercalation and larval sensitivity to noxious mechanical inputs. Altogether, these findings demonstrate that varying the extent of epidermal C4da dendrite intercalation yields corresponding changes in mechanical nociceptive sensitivity.

## Epidermal gap junction proteins limit dendrite intercalation

Intercellular junctions form a barrier restricting paracellular permeability between epidermal cells, and we hypothesized that this barrier was compromised in *miR-14* mutants, providing access for dendrites to intercellular space. When we bathed wild-type larval fillets in rhodamine-conjugated dextran beads, dye accumulated on the apical and basal surfaces of epidermal cells but was excluded from the intercellular space of epidermal cells (Fig 7A). In contrast, dextran beads penetrated between epidermal cells in *miR-14* mutants, labeling intercellular spaces that were also infiltrated by C4da dendrites (Fig 7B). We reasoned that this dye exclusion defect likely reflected dysfunction in one or more of the epidermal junctional complexes: AJs, gap junctions (GJs), and/or septate junctions (SJs) (Fig 7C).

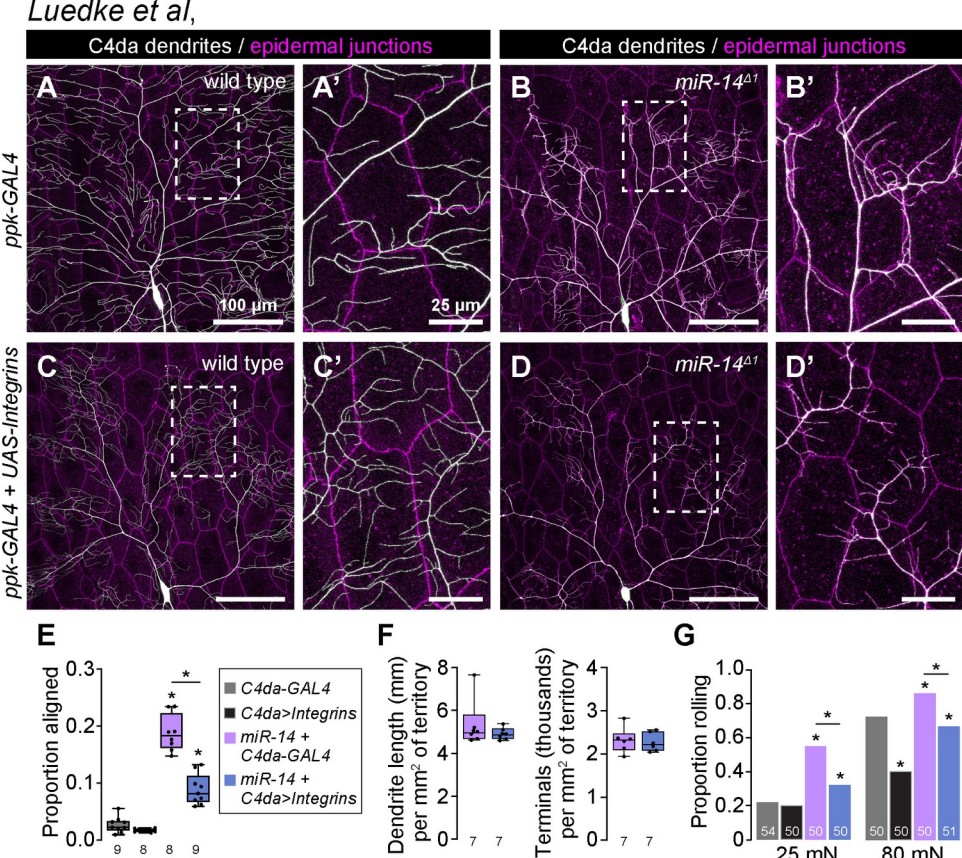

*Luedke et al*,

**Fig 6. Apical epidermal intercalation contributes to mechanical hypersensitivity.** (A-E) Neuronal integrin overexpression suppresses *miR-14* mutant junctional dendrite alignment defect. Representative composite images show C4da dendrite arbors (white) and epidermal cell-cell junctions (magenta) for (A) wild-type control, (B) *miR-14* mutant, (C) control larvae overexpressing *UAS-mew* and *UAS-mys* (*UAS-Integrins*) selectively in C4da neurons (*ppk-GAL4*), and (D) *miR-14* mutant overexpressing *UAS-Integrins* selectively in C4da neurons. Hatched boxes indicate region of interest shown at high magnification to the right of each image. (E) Plot depicts the proportion of dendrite arbors aligned along epidermal junctions in larvae of the indicated genotypes. *P<0.05, ANOVA with post-hoc Tukey's test. (F) Neuronal integrin overexpression has no effect on C4da neuron dendrite branch length or number. Plots depict the total dendrite length (left) and terminal dendrite number (right) normalized to segment size in wild-type control and *miR-14* mutant larvae. Statistical tests (dendrite length, Fisher's exact test; dendrite number, unpaired t-test with Welch's correction) revealed no significant difference in either paramter (n = 7 neurons each). (G) Neuronal integrin overexpression suppresses *miR-14* mutant mechanical hypersensitivity. Plot depicts nociceptive rolling responses of larvae of the indicated genotypes to different forces of von Frey stimulation. *P<0.05, Fisher's exact test with a post-hoc BH correction. The number of larvae tested is shown for each condition.

To identify junctional components regulated by *miR-14* we queried miRNA target prediction databases [47,48], which revealed only one putative junctional *miR-14* target, *Innexin-3* (*Inx-3*). To more directly evaluate *miR-14* regulation of cell junction assembly, we assayed for effects of *miR-14* in epidermal cells on expression of junction-associated genes using RNA-seq analysis (S8A Fig and S1 Table). However, our studies revealed that neither *Inx-3* nor any other junctional components were significantly deregulated in *miR-14* mutant epidermal cells, suggesting that *miR-14* effects on junctional integrity may be indirect.

We therefore tested for genetic interactions between *miR-14* and genes encoding epidermal junction components in control of C4da dendrite morphogenesis and nociceptive sensitivity (Fig 7C). Trans-heterozygous combinations of *miR-14* and mutations in AJ and SJ components had minimal impact on dendrite morphogenesis (S8B Fig), and we found that the levels

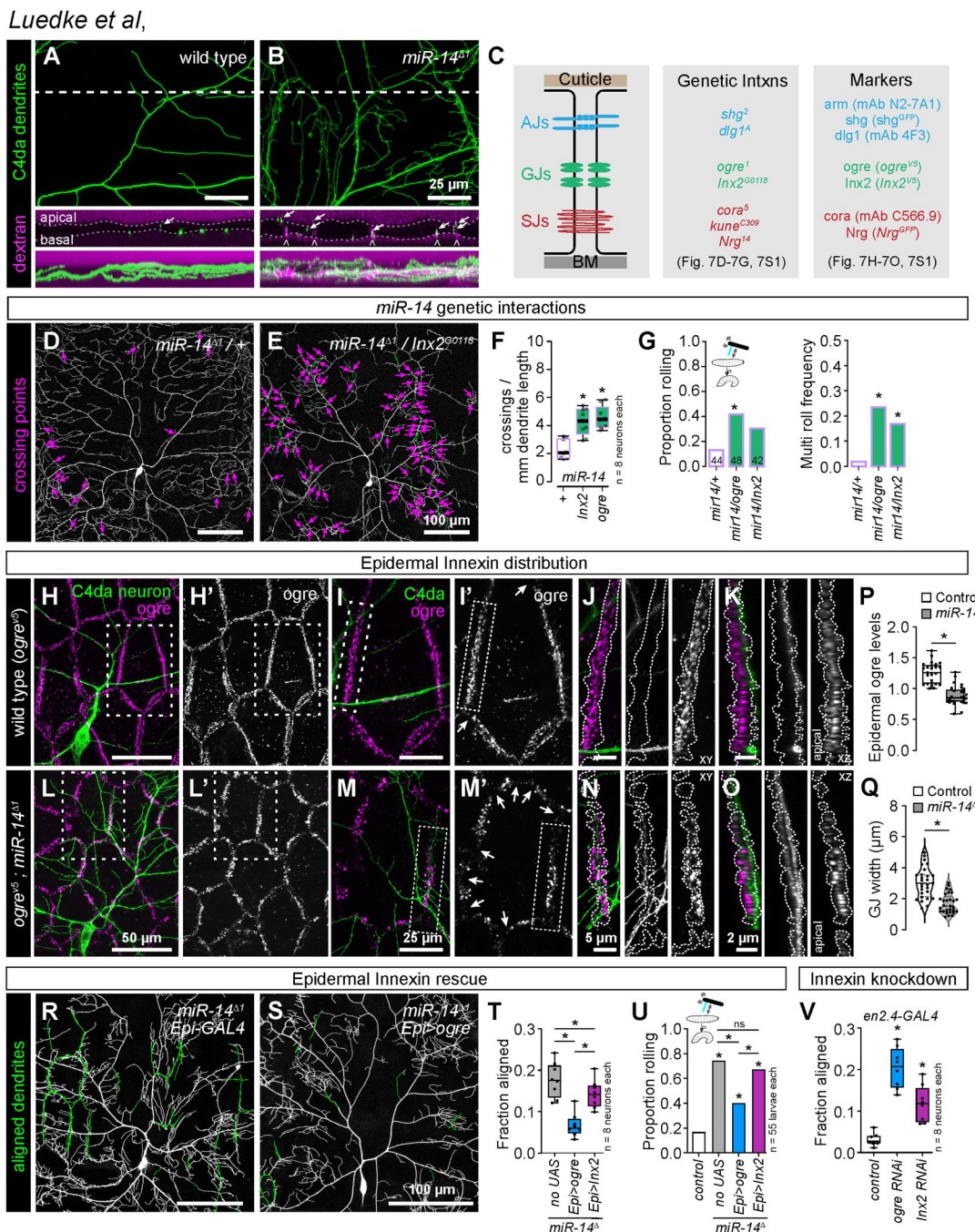

*Luedke et al,*

**Fig 7. *miR-14* regulation of epidermal gap junction assembly controls dendrite position and nociceptive sensitivity.** (A-B) *miR-14* mutation affects epidermal barrier function. Maximum intensity projections show C4da arbors (*ppk-CD4-tdGFP*) and rhodamine-conjugated dextran labeling in cross-section of wild-type control and *miR-14* mutant larvae. Dashed lines indicate the position of the orthogonal xz sections (middle), and bottom images show xz maximum intensity projections. Arrows indicate apically-shifted dendrite branches and carets mark apical dextran infiltration at cell-cell junctions. (C) Schematic depicting position of epidermal junctional complexes (left), alleles used for genetic interaction studies (center), and markers used for analysis of junctional assembly (right). (D-F) *ogre* and *Inx2* genetically interact with *miR-14* to regulate dendrite position. Representative images show 120 h AEL C4da neurons from (D) *miR-14^Δ1^/+* heterozygous mutant and (E) *miR-14^Δ1^/+*, *Inx2^G0118^/+* double heterozygous mutant larvae. (F) Morphometric analysis of C4da dendrites in larvae heterozygous for *miR-14* and the indicated epidermal junction genes showing the mean number of dendrite-dendrite crossing events per neuron normalized to dendrite length. *P<0.05, ANOVA with post-hoc Dunnett's test. (G) *ogre* and *Inx2* genetically interact with *miR-14* to regulate mechanical nociceptive sensitivity. Plots depict the rolling probability and frequency of multiple roll responses evoked by 25 mN von Frey fiber stimulation in larvae of the indicated genotypes. *P<0.05 compared to *miR-14* heterozygous

controls, Fisher's exact test with a post-hoc BH correction (G). (H-O) *miR-14* regulates GJ assembly. Maximum projection images show C4da dendrites (green) and ogre immunoreactivity in the epidermis of a wild-type control larva (H). (I) Zoomed images corresponding to hatched box in (H) show the relative position of C4da dendrites and ogre at epidermal cell-cell junctions. Arrows mark sites of disconiuties in junctional ogre immunoreactivity, which most frequently occurs at tricellular junctions (I').Orthogonal sections show ogre distribution at a representative bicellular junction (outlined with hatched lines) in xy (J) or xz projections (K). C4da dendrites are confined to the basal face of a continuous belt of ogre immunoreactivity in control larvae. (L-O) *miR-14* mutation disrupts organization of ogre immunoreactivity at epidermal cell-cell junctions. (L-M) The belt of ogre immunoreactivity exhibits irregularity in width, signal intensity, and frequent discontinuities (arrows). (N, O). C4da dendrites intercalate into gaps in ogre and immunoreactivity and penetrate apically into the GJ domain. (P-R) Selective epidermal overexpression of *Inx* genes suppresses *miR-14* mutant dendrite alignment and mechanonociception defects. (P) ogre immunoreactivity at GJs is reduced in *miR-14* mutants. Plot depicts intensity of ogre immunoreactivity signal normalized to arm immunoreactivity at epidermal cell-cell interfaces in control and *miR-14* mutant larvae. P<0.05, unpaired t-test with Welch's correction. (Q) GJ belt width is reduced in *miR-14* mutants. Violin plot depicts the distribution of GJ belt widths at epidermal cell-cell interfaces in control and *miR-14* mutant larvae Each data point represents the average GJ belt width measured across the full length of a cell-cell interface. *P<0.05, Kolmogorov-Smirnov test. (R) Composite images show C4da dendrites pseudocolored green to label epidermal junctional alignment in a *miR-14* mutant (left) and a *mir-14* mutant expressing *UAS-ogre* selectively in epidermal cells (right). (R) Plot depicts the fraction of C4da dendrite arbors aligned along epidermal junctions at 120 h AEL for the indicated genotypes. NS, not significant, *P<0.05, Kruskal-Wallis test followed by Wilcoxon rank sum test with BH correction. (S) Fraction of larvae of the indicated genotypes that exhibit nocifensive rolling responses to 25 mN von Frey stimulation. NS, not significant, *P<0.05, Fisher's exact test with a BH correction. (T) Epidermis-specific *Inx* gene knockdown increases epidermal junctional alignment of C4da dendrites. The plot depicts the proportion of C4da dendrite arbors aligned along epidermal junctions in larvae expressing the indicated RNAi transgenes. Quantitative analysis of Inx protein knockdown and representative images of dendrite phenotypes are shown in S9 Fig. *P<0.05, Kruskal-Wallis test followed Dunn's multiple comparisons test.

and distribution of AJ (*shg*$^{RFP}$, Fig 6A and 6B; *arm*$^{GFP}$ and *dlg1*$^{GFP}$, S8C Fig) and SJ markers (cora, *Nrg*$^{GFP}$, S8C Fig) were comparable in *miR-14* mutant and control larvae. In contrast, heterozygous combination of mutations in *miR-14* and genes encoding GJ components yielded synthetic dendrite morphogenesis phenotypes that included a significant increase in dendrite-dendrite crossing (Figs 7D,7F and S8B). Heterozygous mutations in GJ genes likewise yielded synthetic mechanonociceptive sensitization phenotypes in combination with *miR-14* mutation (Fig 7G), suggesting that *miR-14* and GJ genes function together in a genetic pathway to control C4da dendrite position and mechanical nociceptive sensitivity.

*Drosophila* GJs are comprised of homo- or heteromeric assemblies of Inx proteins, eight of which are encoded in the *Drosophila* genome. Our RNA-seq analysis revealed that larval *Drosophila* epidermal cells control and *miR-14* mutant larvae primarily express 3 *Inx* genes: *ogre*, *Inx2*, and *Inx3* (S8A Fig). Among these, only Inx2 appears to form homomeric channels, whereas ogre and Inx3 form heteromeric channels with Inx2 [49,50]. Both ogre and Inx2 function in cell adhesion independent of channel activity [51], and previously described functions for Inx3 in epithelial cells involve Inx2 [49,52]. We therefore focused our analysis on ogre and Inx2, using HA-tagged knock-in alleles to visualize endogenous distribution of these proteins [53].

In control larvae, ogre and Inx2 coalesce into a belt lining epidermal cell-cell interfaces (Figs 7H–7K and S8D), and C4da dendrites are confined to the basal face of this belt of GJ proteins (Fig 7K). The GJ belt occasionally thinned at tricellular junctions (arrows, Fig 7I'), but we rarely observed discontinuities in the GJ immunoreactivity. In contrast, we noted several irregularities in the GJ belt in *miR-14* mutant epidermal cells (Figs 7L–7O and S8E). First, the width and depth of the GJ belt were variable in *miR-14* mutants. On average, the width of the GJ belt was reduced by 44.2% in *miR-14* mutant epidermal cells (Fig 7P). Second, the intensity of GJ immunoreactivity varied across cells and within cell-cell interfaces, with the mean intensity of ogre immunoreactivity within the GJ belt reduced by 29.5% in *miR-14* mutants (Fig 7Q). Third, we observed frequent breaks in the GJ belt (arrows, Fig 7M), and C4da dendrites penetrated these breakpoints, resulting in dendrite invasion into apical domains (Fig 7O). These observations support a model in which GJ proteins restrict dendrite access to intercellular epidermal domains and defects in GJ integrity allow epidermal dendrite intercalation in

*miR-14* mutants. Intriguingly, both *ogre* and *Inx2* were expressed at significantly lower levels at apodeme-apodeme cell interfaces compared to interfaces between other epidermal cells, suggesting that GJ proteins may likewise influence dendrite positioning around apodemes (S8F–S8I Fig).

To test the epistatic relationship between *miR-14* and GJ genes in control of C4da dendrite development we examined whether resupplying GJ gene expression (*UAS-ogre* or *UAS-Inx2*) selectively to epidermal cells mitigated *miR-14* mutant dendrite intercalation and nociceptive sensitivity phenotypes. We found that epidermal expression of *UAS-ogre* or *UAS-Inx2* significantly reduced epidermal dendrite intercalation in *miR-14* mutants, with *UAS-ogre* expression yielding levels of dendrite intercalation comparable to wild-type controls (Fig 7R–7T). Similarly, epidermal *UAS-ogre* and to a lesser degree *UAS-Inx2* expression significantly ameliorated *miR-14* mutant mechanical nociceptive hypersensitivity (Fig 7U), consistent with GJ genes functioning downstream of *miR-14* to limit epidermal dendrite intercalation and nociceptive sensitivity. Neither treatment restored response rates to wild-type levels (Fig 7U), possibly reflecting a shared requirement for ogre and Inx2 or requirements for additional Inx genes. We attempted to test the former possibility by co-expressing *UAS-ogre* and *UAS-Inx2* in a *miR-14* mutant background but were unable to recover viable progeny.

We next sought to evaluate requirements for GJ proteins in limiting epidermal dendrite intercalation. Homozygous loss-of-function *ogre* and *Inx2* mutations are lethal [54,55], and *Inx2* mutation affects epithelial polarity in the embryonic epidermis [56], therefore we used *en-GAL4* in combination with GJ *UAS-RNAi* transgenes to generate epidermal mosaics in which GJ protein levels were reduced in a subset of post-mitotic epidermal cells (S9A–S9E Fig). We found that epidermal knockdown of GJ genes, particularly *ogre*, induced a significant increase in junctional dendrite alignment (Figs 7V and S9F–S9H), suggesting that epidermal GJ proteins are required to limit epidermal dendrite intercalation. We note that *UAS-Inx2-RNAi* expression yielded only a ∼40% knockdown of Inx2, and this may account for the intermediate effects on dendrite intercalation compared to *UAS-ogre-RNAi* expression (S9 Fig). Furthermore, *UAS-ogre-RNAi* expression, which yielded a ∼80% knockdown of ogre, triggered substantial epidermal cell loss, suggesting that epidermal cells are sensitive to high-level knockdown of GJ genes. Altogether, these results demonstrate that GJ proteins are required in epidermal cells to limit dendrite intercalation and suggest that *miR-14* limits epidermal dendrite intercalation via control of epidermal GJ assembly.

## Epidermal dendrite intercalation tunes mechanical sensitivity of C4da neurons

Our results support a model in which the extent of epidermal dendrite intercalation tunes mechanical sensitivity of nociceptive C4da neurons. To test this model, we expressed the calcium indicator *UAS-GCaMP6s* selectively in C4da neurons and monitored effects of *miR-14* mutation on mechanically evoked calcium responses (Fig 8A). Although we observed no significant difference in the amplitude of response in *miR-14* mutants ($\Delta F/F_0$, Fig 8B and 8C), GCaMP6s fluorescence intensity was significantly elevated in *miR-14* mutants, both before and after mechanical stimulus (Fig 8B and 8D), suggesting that *miR-14* affects C4da baseline calcium levels in our imaging preparation. To corroborate these results, we selectively expressed a GCaMP6m-Cherry fusion protein (*UAS-Gerry*) in C4da neurons and used ratiometric imaging to monitor calcium levels in unstimulated larvae [57]. Indeed, we found that the GCaMP/mCherry ratio, a proxy for baseline calcium, was elevated in *miR-14* mutant C4da neurons (Fig 8D–8F). Furthermore, this increase in the GCaMP/mCherry ratio was substantially suppressed by C4da-specific integrin overexpression, which blocks epidermal dendrite

*Luedke et al,*

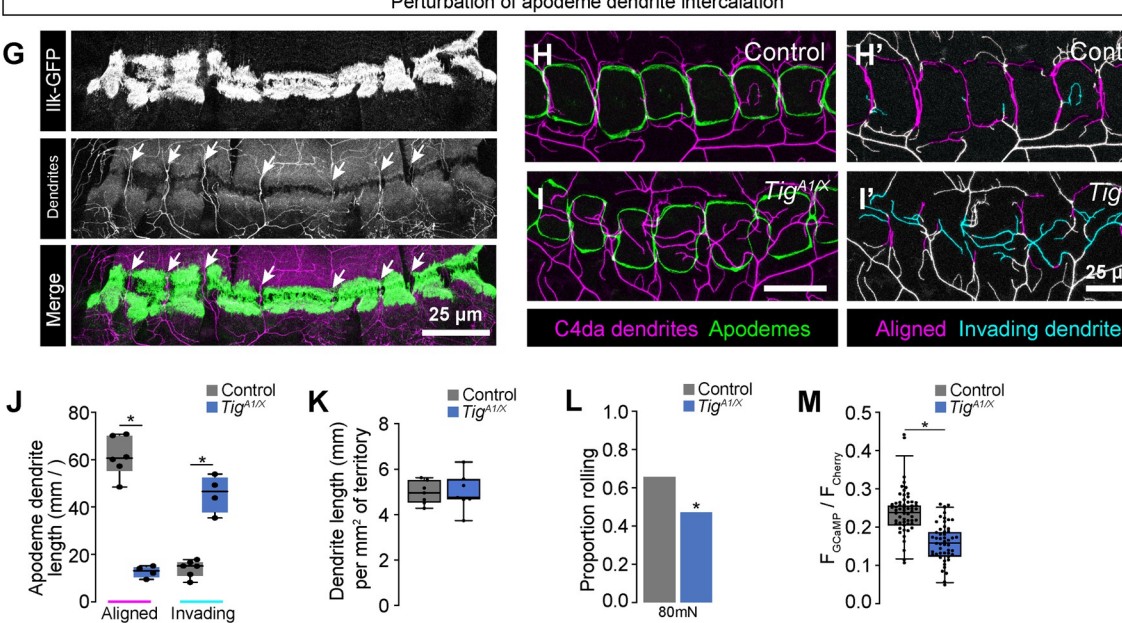

**Fig 8. Epidermal dendrite intercalation tunes C4da neuron calcium levels and nociceptive sensitivity.** (A-F) Epidermal dendrite intercalation regulates baseline calcium levels in C4da neurons. (A) Larval preparation for imaging mechanically-evoked calcium responses in C4da neurons. (B) Control and *miR-14* mutant larvae exhibit comparable amplitudes of GCaMP6s responses ($\Delta F/F_0$) in C4da neurons to mechanical stimulus ($n$ = 15 larvae each, $p$ = 0.217, Wilcoxan rank sum test). (C) GCaMP6s fluorescence intensity is significantly elevated in *miR-14* mutant C4da axons prior to mechanical stimulus and 5 min after mechanical stimulus ($n$ = 15 larvae each, $p$ = 0.0003 pre-stimulus, $p$ = 0.002 post-stimulus, Wilcoxon rank sum test). (E-F) Ratiometric calcium imaging using a GCaMP6s-Cherry fusion protein expressed selectively in C4da neurons (*ppk-GAL4, UAS-Gerry*). (E) Representative images depict fluorescence intensity of Cherry and GCaMP6s for wild-type control and *miR-14* mutant larvae. (F) *miR-14* mutants exhibit elevated GCaMP/mCherry fluorescence ratios in C4da axons. Points represent measurements from individual abdominal segments (A2-A8) from 10 larvae of each genotype (n = 64 data points in control larvae, 56 in *miR-14* mutants, 66 in *miR-14* + *ppk>Integrin* larvae). *P<0.05, Kruskal-Wallis test followed by Dunn's post-hoc test. (G) C4da dendrites are confined to territory between muscle adhesion sites at apodemes. (H-J) Mutation of the PS2 integrin ligand *Tig* prevents dendrite intercalation between apodemes. Representative maximum intensity projections of larvae expressing *ppk-CD4-tdTomato* to label C4da dendrites and *Nrg^GFP* to label epidermal junctions are shown for (H) *Tig^{A1/+}* heterozygote control and (I) *Tig^{A1/X}* mutant larvae. C4da dendrites are depicted in magenta and apodemes in green using an apodeme mask to subtract *Nrg^GFP* signal in other cells. (H' and I') C4da dendrites are pseudocolored according to their orientation at apodemes (magenta, aligned along apodeme junctional domains; cyan, invading apodeme territory). (J) *Tig^{A1/X}* mutant larvae exhibit a significant reduction in junctional dendrite alignment at apodemes and an increase in dendrite invasion into apodeme territory. *P<0.05, ANOVA with a post-hoc Sidak's test. (K) *Tig^{A1/X}* mutant larvae exhibit no significant alteration in dendrite arbor length outside of apodeme domains. No significant difference was detected between the two groups using an unpaired t-test. (L-M) Dendrite intercalation at apodemes tunes nociceptive sensitivity. (L) *Tig^{A1/X}* mutant larvae exhibit a significant reduction in rolling responses to noxious mechanical stimulus. (M) *Tig^{A1/X}* mutant larvae exhibit a significant reduction in GCaMP/mCherry fluorescence ratios in C4da axons Points represent measurements from individual neurons. *P<0.05, Wilcoxon rank sum test in (L-M).

intercalation (Fig 8F). These results are consistent with the model that epidermal intercalation contributes to *miR-14*-dependent elevation of C4da neuron baseline calcium levels and hence modulates mechanical sensitivity of C4da neurons.

Apodemes are the primary site of epidermal dendrite intercalation in wild-type larvae, therefore in a final line of experiments we investigated the contribution of apodeme-dendrite interactions to mechanical nociceptive sensitivity. Apodemes serve as muscle attachment sites in the body wall, and simultaneous labeling of muscles with *Mhc*::*Cherry* [58], muscle attachment sites (*Ilk-GFP*) [33], and C4da dendrites (*ppk-CD4-tdTomato*) revealed that sites of C4da dendrite intercalation between apodemes occurs at gaps between muscle attachments, suggesting that dendrite exclusion from muscle adhesion sites likely contributes to dendrite orientation over apodemes (Fig 8G). We therefore assayed effects on apodeme dendrite coverage of compromising muscle attachments, making use of hypomorphic mutations in *Tiggrin* (*Tig*), which encodes an apodeme-enriched PS2 integrin ligand that functions to maintain muscle attachment [59]. We found that *Tig* mutation altered dendrite distribution at apodemes, significantly reducing the extent of dendrite intercalation between apodemes and increasing dendrite invasion into apodeme territory (Fig 8H–8J) without affecting dendrite arborization patterns outside of apodeme domains (Fig 8K). In addition to these morphogenetic defects, *Tig* mutation significantly reduced behavioral responses to noxious mechanical inputs (Fig 8N) as well as C4da calcium levels in unstimulated larvae (Fig 8O), suggesting that dendrite insertion between apodemes contributes to nociceptive sensitivity. Altogether, our results demonstrate that dendrite-epidermal interactions shape responses to nociceptive inputs, with the extent of dendrite intercalation between epidermal cells tuning nociceptor mechanical sensitivity.

## Discussion

An animal's skin is innervated by a diverse array of SSNs that exhibit type-specific arborization patterns and response properties, both of which are subject to regulation by epidermal cells. Despite evidence that SSNs differentially interact with distinct populations of epidermal cells, contributions of epidermal diversity are an underappreciated determinant of SSN patterning. Here, we identified a genetic pathway that controls the position of nociceptive SSN dendrites in the epidermis and hence sensitivity to noxious mechanical cues. Specifically, we found that the miRNA *miR-14* regulates the levels and distribution of GJ proteins to restrict intercalation of nociceptive C4da dendrites into epidermal junctional domains. This pathway has two notable axes of specificity. First, *miR-14* regulates dendrite intercalation at epidermal but not apodeme cell interfaces, consistent with its expression pattern. Second, *miR-14* regulates dendrite positioning of nociceptive C4da neurons but no other SSNs; *miR-14* likewise controls larval responses to noxious but not non-noxious mechanical cues. Our studies therefore establish a role for nociceptor-specific epidermal interactions in tuning nociceptor response properties in *Drosophila* and more broadly suggest that sensitivity to mechanical nociceptive cues is subject to epidermal control.

Many cutaneous receptors form specialized structures with epidermal cells or other skin cells that contribute to somatosensation [60]. Low threshold mechanoreceptor (LTMRs) form synapse-like contacts with Merkel cells [61], which respond to mechanical stress and tune gentle touch responses by releasing excitatory neurotransmitters that drive static firing of LTMRs [62,63]. Similarly, several types of mechanoreceptors form specialized end organs with accessory cells that shape transduction events [64]. For example, epidermal cells in adult *Drosophila* ensheathe the base of mechanosensory bristles and amplify touch-evoked responses [65]. Afferent interactions with Schwann cell-derived lamellar cells facilitate high frequency

sensitivity in Pacinian corpuscles [66], and different populations of Meissner corpuscles have distinctive lamellar wrapping patterns that may dictate their response properties [67]. While less is known about epidermal control of nociception, the finding that epidermal ensheathment influences sensitivity to noxious mechanical cues provides one mechanism by which epidermal interactions modulate nociception [5]. Our study defines a role for another type of epidermis-nociceptor interaction, epidermal dendrite intercalation, in controlling mechanically evoked responses of nociceptive neurons. Genetic manipulations that triggered widespread dendrite intercalation, including loss-of-function mutations in *Dcr* and *miR-14*, enhanced nocifensive responses to harsh mechanical stimuli without affecting responses to non-noxious mechanical cues. Furthermore, genetic treatments that suppressed or enhanced *miR-14* mutant epidermal dendrite intercalation phenotypes had corresponding effects on nociceptive sensitivity. In contrast, mutations in *Tig*, which eliminated dendrite intercalation between apodemes, reduced nocifensive behavioral responses to mechanical inputs.

We identified three key players in control of mechanical nociception by epidermal dendrite intercalation: the miRNA *miR-14*, epidermal GJ proteins, and the mechanosensory channel Piezo. Among these, *miR-14* is the specificity factor that determines where intercalation occurs: dendrite intercalation is largely restricted to apodeme domains in wild-type larvae by selective epidermal expression of *miR-14* outside of apodeme domains. In the simplest model, *miR-14* would function as a molecular switch to toggle between apodeme and epidermal cell fate and hence control dendrite intercalation. However, our expression studies demonstrate that *miR-14* does not regulate epidermal expression of apodeme marker genes or regulators of apodeme fate, and ectopic *miR-14* expression had no obvious effect on apodeme morphology or position. Instead, our studies support a model in which *miR-14* regulates intercellular interactions between epidermal cells to control dendrite access to junctional domains; in addition, exclusionary interactions between muscle adhesions and dendrites likely contribute to the extensive dendrite intercalation at apodemes. Although *miR-14* orthologues are not readily apparent in vertebrates, numerous miRNAs are expressed in discrete subsets of epidermal cells and regulate multiple aspects of skin development including epidermal barrier formation [68, 69]. Hence, miRNAs may likewise function as specificity factors that dictate the position of SSN neurites in vertebrate skin.

In other developmental contexts, *miR-14* directly targets Hedgehog signaling pathway components [70], the *IP3 kinase* to influence IP3 signaling [71], and the transcription factors *sugarbabe* and *Ecdysone receptor* to regulate Insulin production and steroid signaling, respectively [34,72]. And although both Hedgehog and Insulin signaling affect injury-induced hypersensitivity to noxious heat [73,74], *miR-14* mutation specifically sensitizes larvae to noxious mechanical cues. We found no evidence that previously defined *miR-14* targets control dendrite-epidermis interactions. Instead, our studies suggest that *miR-14* control of dendrite intercalation and mechanical nociceptive sensitivity is gated by GJs: *miR-14* mutants exhibit reduced epidermal Inx expression (ogre and Inx2) and discontinuities in the epidermal belt of GJ proteins. Furthermore, epidermal intercalation in wild-type larvae principally occurs between apodemes, which exhibit reduced levels of *miR-14*, ogre, and Inx2 compared to other epidermal cells. How do GJ proteins regulate dendrite accessibility to epidermal junctional domains? In addition to their roles in GJ or hemi-channel transport, ogre and Inx2 act as adaptor proteins independent of channel function to regulate intercellular interactions in several contexts. For example, Inx2 acts upstream of integrin-based adhesion to promote follicle cell flattening during *Drosophila* ovary morphogenesis [75], and both Inx2 and ogre play channel-independent adhesive roles in coupling subperineurial and wrapping glia [51]. Within the embryonic epidermis, Inx2 interacts with SJ and AJ proteins to control localization of junctional proteins, including E-cadherin, and epithelial polarization [52,76], but postembryonic

epidermal Inx functions have not been defined. Our results are consistent with two possible models. First, gap junction proteins may regulate localization of a factor that actively prevents dendrite intercalation at epidermal junctions; such a factor could repel dendrites via short-range interactions. Second, gap junctions and associated factors could physically occlude intercellular space and hence prevent dendrite access. Our finding that epidermal junctions in *miR-14* mutants exhibit heightened permeability to dextran beads is consistent with the latter possibility.

In vertebrates, GJs mediate intercellular communication in sensory ganglia [77] and injury-induced upregulation of GJs can facilitate mechanical hyperalgesia through heightened cross-depolarization [78]. Whether GJs additionally function in the periphery to control vertebrate mechanonociception is not known, however several studies point towards such a function. Different epidermal cell types express unique combinations of GJ proteins with different permeability properties [79,80], and this may contribute to layer-specific or regional differences in neuronal coupling to epidermal cells. GJ gating can be modulated by mechanical, thermal and chemical stimuli [81,82], providing a potential mechanism for epidermal integration of sensory information. Indeed, studies in *ex vivo* preparations suggest that GJs mediate keratinocyte ATP release in certain circumstances including mechanical injury [82,83], and that keratinocyte ATP release can drive nociceptor activation [84]. Finally, GJs serve channel-independent functions in the skin: connexin mutations are linked to inflammatory skin disorders as well as diseases that affect epidermal thickness and barrier function [85] and GJ blocking agents inhibit the barrier function of tight junctions in cells [86].

Mechanistically, how could epidermal intercalation influence mechanical nociception? First, insertion into the apical junctional domain could expose dendrites to a different extracellular environment that locally influences gating properties of sensory channels. Although epidermal junctions in *miR-14* mutants exhibited increased permeability to dendrites, dextran dyes, and likely extracellular solutes, neuronal excitability could be influenced through interactions with extracellular domains of proteins enriched in apical junctional domains. Second, epidermal dendrite intercalation could facilitate nociceptor activation through enhanced ionic coupling to epidermal cells, analogous to lateral excitation that improves motion sensitivity of ON bipolar cells [87]. In such a case, nociceptors would exhibit heightened sensitivity to epidermal ion flux, and recent studies indicate that epidermal stimulation influences response properties of SSNs including *Drosophila* nociceptors [88–90]. However, we found that mutation of the mechanosensory channel gene *Piezo* decoupled dendrite intercalation and mechanical hypersensitivity. Hence, we favor a third possibility, namely that dendrite intercalation influences mechanosensory ion channel function in nociceptive neurons.

Epidermal junctions are under constant tensile stress and subject to a range of additional forces including those generated during locomotion and by local compression of the skin. Dendrites that insert into junctional domains may therefore experience heightened tensile stress in the absence of noxious inputs and enhanced forces in response to mechanical stimuli. Such a model is particularly appealing given our observation that the mechanical sensitization depends on *Piezo* and recent studies indicating that Piezo channels are gated by changes in lateral membrane tension [91–93]. Mechanical perturbations of the lipid bilayer alone are sufficient to activate Piezo channels [94], with membrane deformation likely bending the blades of the channel to gate the pore [95,96]. Within C4da dendritic arbors, Piezo mediates responses to localized forces [97]; we propose that Piezo is likewise responsive to localized membrane stress transduced at intercellular junctions. In wild-type larvae, dendrite intercalation principally occurs at sites of body wall muscle attachment, therefore dendrite intercalation at these sites could dynamically couple Piezo channel activity to locomotion. In *miR-14 mutants*, C4da dendrites inappropriately intercalate between epidermal cells outside of apodeme cells, and

our studies suggest that this increased dendrite-epidermis coupling enhances nociceptive sensitivity. It remains to be determined whether the low level of intercalation that normally occurs outside of apodeme domains ($\sim$4% of the arbor; Fig 1G) contributes to nociceptive responses in wild-type larvae. More broadly, our studies suggest that the extent and/or depth of nociceptor intercalation within keratinocyte layers could influence mechanical response properties of vertebrate DRG neurons in both physiological and pathological states. Several skin disorders associated with barrier dysfunction including atopic dermatitis exhibit enhanced sensitivity to mechanical inputs, therefore it will be intriguing to determine whether inappropriate apical neurite invasion and epidermal intercalation contribute to the mechanical hypersensitivity seen in these disorders.

## Materials and methods

### Genetics

**Drosophila strains.** Flies were maintained on standard cornmeal-molasses-agar media and reared at 25°C under 12 h alternating light-dark cycles. See S2 Table for a complete list of alleles used in this study; experimental genotypes are listed in figure legends.

**EMS mutagenesis.** *Mutations affecting dendrite position*: EMS mutagenesis was previously described [24]. The $Dcr^{dg29}$ allele carries a single nucleotide change (G5711T) that results in an amino acid substitution (G1905V) adjacent to the catalytic site.

*miR-14 modifier screen*: $miR\text{-}14^{\Delta1}$, $ppk\text{-}CD4\text{-}tdGFP^{1b}$ was outcrossed for 4 generates to $w^{1118}$, balanced over *Cyo-Tb*, and mutagenized with EMS as above. Balanced stocks were established from $\sim$600 $miR\text{-}14^{\Delta1}$, $ppk\text{-}CD4\text{-}tdGFP^{1b}$, *EMS\** F2 single males, and 20 homozygous mutant larvae from each of these mutant lines were screened for mechanonociceptive responses to 25 mN von Frey stimulus. Mutant lines with z-scores (absolute values) > 2 were retained and rescreened 2x, with additional filtering each round. Finally, candidate suppressors and enhancers (8 total) were subjected to mechanonociception assays (>40 homozygous mutant larvae each), and mutants with response rates that were significantly enhanced or reduced in comparison to $miR\text{-}14^{\Delta1}$, $ppk\text{-}CD4\text{-}tdGFP^{1b}$ were retained and subjected to further analysis. The screen yielded one mutant allele ($mda1^{246}$) that significantly enhanced *miR-14* nocifensive responses to mechanical stimulus. Three additional alleles which had more variable effects on *miR-14* responses (two putative enhancers, one putative suppressor) were retained for further analysis.

**miR-14-GAL4.** The *miR-14-GAL4* driver was generated by ligating a 3 kb promoter fragment (forward primer: gaagctagctcgaccccatggtgtagg; reverse primer: gaaggatcctaggttgcagtacgttacgtt) digested with NheI and BamHI into pPTGAL4 digested with XbaI and BamHI. Injection services were provided by Bestgene.

### Microscopy

**Live confocal imaging.** Live single larvae were mounted in 90% glycerol under a coverslip and imaged on a Leica SP5 confocal microscope using a 20x 0.8 NA or 40x 1.25 NA lens. Larvae subject to time-lapse microscopy were recovered between imaging sessions to plates containing standard cornmeal-molasses-agar media. For high-resolution confocal imaging (Figs 3 and 7), image stacks were acquired using the following acquisition settings to ensure Nyquist sampling: 1024 x 1024 pixels, 4x optical zoom (96.875 x 96.875 μm field of view), 150–300 nm optical sections.

**Calcium imaging.** *Mechanically-evoked responses*: Third-instar larvae were pinned on sylgard (Dow Corning) dishes with the dorsal side up bathed in calcium-containing HL3.1 [98].

40mN mechanical stimuli were delivered between segments A2-A4 using a calibrated von Frey filament and images were captured immediately prior to and following mechanical stimulus.

*Calcium responses during peristalsis*: Intact third-instar larvae were immobilized dorsal side up on sylgard plates, loosely pinned to limit longitudinal stretch and to allow peristaltic movements and were bathed in calcium-containing HL3.1. Larvae were acclimated to the imaging arena for 2 min and calcium responses of C4da neurons (Gerry GCaMP6m fluorescence) were captured during peristaltic movement for 30 sec under constant illumination at a frame acquisition rate of 10 fps.

**Immunostaining.** Third instar larvae were pinned on a sylgard plate, filleted along the ventral midline, and pinned open. After removing intestines, fat bodies, imaginal discs and ventral nerve cord, fillets were fixed in PBS with 4% PFA for 15 min at room temperature, washed 4 times for 5 min each in PBS with 0.3% Tx-100 (PBS-Tx), blocked for 1 h in PBS-Tx + 5% normal goat serum, and incubated in primary antibody overnight at 4° C. Samples were washed 4 times for 5 min each in PBS-Tx, incubated in secondary antibody for 4 h at room temperature, washed 4 times for 5 min each in PBS-Tx, and stored in PBS prior to imaging. Antibody dilutions were as follows: chicken anti-GFP (Aves Labs #GFP-1020, 1:500), mouse anti-coracle (DSHB, C566.9 supernatant, 1:25), mouse anti-armadillo (DSHB, N2-7A1 supernatant, 1:25), mouse anti-discs large (DSHB, 4F3 supernatant, 1:25), Rabbit anti-V5 (Biolegend #903801, 1:500), HRP-Cy5 (Jackson Immunoresearch, 1:100), Goat anti-Mouse Alexa488 (Thermofisher A-11001, 1:200), Goat anti-Chicken Alexa488 (Thermofisher A-11039, 1:200), Goat anti-rabbit Alexa 488 (Thermofisher A-11034, 1:200), Goat anti-Mouse Alexa555 (Thermofisher A-28180, 1:200), Goat anti-rabbit Alexa555 (Thermofisher A-21428, 1:200).

## Image analysis

**Morphometric analysis of C4da dendrites (Figs 1, 4, 6–8, S1, S2 and S4).** Features of dendrite morphology, including dendrite branch number, dendrite length, and dendrite crossing number were measured in Fiji [99] using the Simple Neurite Tracer plugin [100]. Epidermal junction alignment and ensheathment of dendrites was measured from manual traces generated from composite images (neuronal marker + epidermal junction or sheath marker) in Fiji. Ensheathed stretches [5] and junction-aligned stretches were identified via co-localization with the epidermal PIP2 marker PLC$^\delta$-PH-GFP and manually traced in Fiji. Dendrite stretches that tracked junctional PLC$^\delta$-PH-GFP labeling for > 2 μM (approximately 2x the mean width of junctional domains) were considered aligned. To monitor dendrite-apodeme interactions *Nrx-IV^GFP* signal was used to generate an apodeme mask, and *ppk-CD4-tdTomato* signal aligned to apodeme borders (apodeme-aligned dendrites) and contained within the apodeme mask (invading dendrites) was manually traced in Fiji and normalized to apodeme territory.

**Analysis of time-lapse images (Figs 2 and S3).** Epidermal junction-aligned dendrites were identified as stretches of C4da dendrites overlapping with *shg^Cherry* signal in maximum projections of confocal stacks and measured via manual tracing in Fiji [99]. Corresponding regions of interest from early and late timepoints were registered to one another and dynamics of junction-aligned dendrites were measured in image composites. To measure the orientation of new branch growth, normal lines were drawn to lines connecting tricellular junctions at the vertex of each epidermal cell-cell interface, and angles of incidence between junction-crossing dendrites and the normal lines were measured using the ImageJ angle plugin.

**Axial positioning of C4da dendrites (Fig 3).** *ECM detachment*: Image stacks were captured at a Z-depth of 150 nm and deconvolved using the Leica LAS deconvolution plugin set to adaptive PSF for 10 iterations. 3D reconstruction was performed with Imaris and co-

localization was measured between fluorescent signals labeling dendrites (*ppk-CD4-tdTomato*) and ECM (*trol^GFP*) using the Imaris Coloc module. The dendrites were traced in Imaris, portions of the arbor that failed to co-localize with the ECM (apically detached dendrites) were pseudocolored in traces, and the proportion of the arbor that was detached from the ECM was measured in these traces.

*Axial distance between dendrites and AJs*: Image stacks were captured at a Z-depth of 250 nm, Z-depths of signal peaks for dendrites and AJs were extracted for each junctional crossing event, and the difference in these two values was calculated as the axial distance. To monitor axial position of aligned dendrites over extended lengths, dendrites were segmented into 500 nm sections and axial distance between dendrites and AJs was measured for each section.

**Epidermal features (Figs 3, 8 and S8).** *Length of epidermal cell-cell interfaces*: Image stacks were captured at a Z-depth of 250 nm and epidermal cells containing intercalated dendrites were identified using the following criteria for dendrite intercalation: dendrite angle of incidence of $> 60°$ with the cell-cell interface, apically displacement of the dendrite to within 1 μm of shg-Cherry signal, and a segment length of $> 2$ μM oriented along the junctional domain. Lengths of cell interfaces were measured by manual tracing of *shg^Cherry* signal spanning the interfaces in maximum projections of image stacks. Edges from 55 cells were traced with no prior knowledge of dendrite intercalation status to generate a population distribution for all junctions, and 28 edges containing intercalated dendrites were measured to generate a population distribution for intercalated junctions.

*Gap junction width*: Width of anti-V5 immunoreactivity (*ogre^V5* signal) was measured at 500 nm intervals along the entire cell-cell interface of epidermal cells to calculate a mean width for that edge. Mean width values were calculated for n = 50 cell-cell interfaces in control and *miR-14* mutants.

*Distance from tricellular junction*: Sites of dendrite intercalation (where basally-localized dendrites penetrate apically at junctional domains) were identified in image stacks and the lateral distance from the insertion site to the nearest tricellular junction was measured by manually tracing shg-Cherry signal in maximum projections of image stacks. Midpoint values were calculated as the half-width of edge lengths (distance between tricellular junctions) from the 30 cell interfaces sampled.

*Mean junctional width of GJ immunoreactivity*: Bands of Inx immunoreactivity at apodeme-apodeme interfaces located at segment boundaries (which are oriented along the AP axis) and at epidermis-epidermis interfaces in neighboring segments were segmented in maximum projections of image stacks. The width of these junctional bands was measured at positions spaced by 100 nm along the entire length of the cell interface, and the mean of these measurements was taken as the mean junctional width of GJ immunoreactivity.

**Reporter intensity (Figs 4, S7 and S9).** *miR-14-GAL4 reporter*: Image stacks were collected under conditions to limit signal saturation. Cell outlines were manually traced in maximum projections of confocal stacks using *Nrg-GFP* signal to mark plasma membranes. Mean RFP intensity values were measured for individual cells and background signal measured from stage-matched *UAS-tdTomato* larvae imaged under the same settings was subtracted to generate expression values. Epidermal reporter values adopted a trimodal distribution: cells that did not express *UAS-tdTomato* above background levels, cells with mean signal intensity below 30 AU, and cells with mean signal intensity above 60 AU. These cells were designated non-expressers, low expressers, and high expressers, respectively.

*miR-14 activity sensor*: Image stacks were collected under conditions to limit signal saturation, using identical settings for all samples. Cell outlines were manually traced in maximum projections of confocal stacks using miRNA sensor GFP signal to mark epidermal plasma membranes and tdTomato signal (*sr-GAL4*) to mark apodeme membranes. Mean GFP

intensity values were measured for individual cells and background signal measured from stage-matched sibling cross-progeny lacking miRNA sensor transgenes was subtracted to generate expression values.

*ogre/inx2 levels at epidermal junctions*: Image stacks were collected under conditions to limit signal saturation, using identical settings for all samples, and mean intensity values for V5 immunoreactivity (ogre or Inx2) were measured from manually drawn ROIs that encompassed individual epidermal or apodeme cell-cell interfaces.

**Calcium imaging ([Fig 8]).** *Mechanically-evoked responses*: ROIs containing cell bodies were manually traced in ImageJ [101] using GCaMP signal as a guide, and ROIs for background subtraction were drawn adjacent to C4da neurons in territory lacking signal from cell bodies or neurites. $\Delta F$ was calculated by the formula F(after)-F(before)/F(before) using background-subtracted measurements.

*Ratiometric imaging*: ROIs containing axon terminals from a single hemisegment were manually traced in ImageJ [101] using mCherry signal in the ventral ganglion as a guide. Mean mCherry ($F_{Cherry}$) and GCaMP6m ($F_{GCaMP}$) signal intensity values were measured for individual hemisegments, background signal was measured outside of the field of view of the specimen, and background-subtracted values were used to calculate fluorescence ratios. Measurements were taken from abdominal segments of 12 larvae for each genotype.

## Behavior assays

**Mechanonociception assays.** Third instar larvae were isolated from their food, washed in distilled water, and placed on a scored 35 mm petri dish with a thin film of water such that larvae stayed moist but did not float. Larvae were stimulated dorsally between segments A4 and A7 with calibrated Von Frey filaments that delivered the indicated force upon buckling, and nocifensive rolling responses were scored during the 10 sec following stimulus removal.

**Thermonociception assays.** Animals were rinsed in distilled water and transferred to a moistened 2% agar plate. Thermal stimuli were delivered to the lateral side of larvae and targeted to segments A5/A6 using a heat probe consisting of a soldering iron modified to include an embedded thermocouple and powered by a voltage transformer. Thermal stimulus was continuously applied to each larva for 10 seconds, each larva was scored as a roller or non-roller, and latency was scored as the time elapsed from stimulus onset until roll initiation. Temperature was listed temperature $\pm$ 0.5˚C.

**Gentle touch assays.** Animals were rinsed in distilled water, transferred to a moistened 2% agar plate, and habituated for 1 minute prior to behavior trials. Larvae were stimulated on anterior segment with an eyelash probe four times (5 sec recovery between each trial) during locomotion for each trial. Behavior responses were scored as previously described [102], and scores for the four trials were summed to calculate Kernan scores.

**Sound/vibration responses.** Wandering third instar larvae were picked from a vial and washed with PBS. 10 larvae were placed on a 1% agar plate on top of a speaker and stimulated as previously described [43]. A 1-second 70dB, 500Hz pure tone was played 10 times with 4 seconds of silence in between. Video recordings captured larval behavior, with the number of times out of 10 each larva exhibited sound startle behavior as its individual score. 3 separate trials were performed for each genotype. Larval startle behavior was scored as responsive with the following behaviors: mouth-hook retraction, pausing, excessive turning, and/or backward locomotion.

**Vinblastine treatment.** At 96 h AEL larvae were transferred to standard cornmeal-molasses-agar food supplemented with vinblastine (10 um) or vehicle (DMSO) as well as green food coloring to monitor food uptake. Following 24 h of feeding, larvae with significant food

uptake, scored by presence of food dye in the intestinal tract, were assayed for nociceptive responses to 25 mN von Frey stimulation.

**UV treatment.** At 96 h AEL larvae were rinsed in distilled water, dried, transferred to 100 mM dishes, and subjected to 20 mJ/cm2 UV irradiation using a Stratalinker 2400 UV light source (Stratagene) as previously described [7]. Following irradiation, larvae were recovered to 35 mM dishes containing standard cornmeal-molasses-agar food for 24 h and subsequently assayed for responses to 25 mN von Frey stimulation or noxious heat.

## RNA-Seq analysis of epidermal cells

**RNA isolation for RNA-Seq.** Control ($w^{1118}$) or *miR-14* mutant ($w^{1118}; miR-14^{A1}$) larvae with cytoplasmic GFP expressed in the cells of interest (apodemes, *UAS-2x-EGFP/+; sr-GAL4/+*; epidermal cells, *UAS-2x-EGFP/+; R38F11-GAL4/+*) were microdissected and dissociated in collagenase type I (Fisher 17-100-017) into single cell suspensions, largely as previously described [103], with the addition of 1% BSA to the dissociation mix. After dissociation, cells were transferred to a new 35 mm petri dish with 1 mL 50% Schneider's media, 50% PBS supplemented with 1% BSA. Under a fluorescent stereoscope, individual fluorescent cells were manually aspirated with a glass pipette into PBS with 0.5% BSA, and then serially transferred until isolated without any additional cellular debris present. Ten cells per sample were aspirated together, transferred to a mini-well containing 3ul lysis solution (0.2% Triton X-100 in water with 2 U / μL RNAse Inhibitor), lysed by pipetting up and down several times, transferred to a microtube, and stored at -80˚ C. For the picked cells, 2.3 μL of lysis solution was used as input for library preparation.

**RNA-Seq library preparation.** RNA-Seq libraries were prepared from the picked cells following the Smart-Seq2 protocol for full length transcriptomes [104]. To minimize batch effects, primers, enzymes, and buffers were all used from the same lots for all libraries. Libraries were multiplexed, pooled, and purified using AMPure XP beads, quality was checked on an Agilent TapeStation, and libraries were sequenced as 51-bp single end reads on a HiSeq4000 at the UCSF Center for Advanced Technology.

**RNA-Seq data analysis.** Reads were demultiplexed with CASAVA (Illumina) and read quality was assessed using FastQC (https://www.bioinformatics.babraham.ac.uk/) and MultiQC [105]. Reads containing adapters were removed using Cutadapt version 2.4 [106] (Martin, 2011) and reads were mapped to the *D. melanogaster* transcriptome, FlyBase genome release 6.29, using Kallisto version 0.46.0 [107] with default parameters. AA samples were removed from further analysis for poor quality, including low read depth ($< 500,000$ reads), and low mapping rates ($< 80\%$). Raw sequencing reads and gene expression estimates are available in the NCBI Sequence Read Archive (SRA) and in the Gene Expression Omnibus (GEO) under accession number GSE262604.

## Experimental design and statistical analysis

For each experimental assay control populations were sampled to estimate appropriate sample numbers to allow detection of $\sim 33\%$ differences in means with 80% power over a 95% confidence interval. Datasets were tested for normality using Shapiro-Wilks goodness-of-fit tests. Details of statistical analysis including treatment groups, sample numbers, statistical tests, controls for multiple comparisons, p-values and q-values are provided (S1 Data).

**Experimental genotypes (Fig 1).** (A-D) $w^{1118};; Nrx-IV^{GFP}, ppk-CD4-tdTomato^{10A}$
(E-K) <u>control</u>: $w^{1118};; A58-GAL4, UAS-PLC^{\delta}-PH-GFP, ppk-CD4-tdTomato^{10A}$
<u>miR-14</u>: $w^{1118}; miR-14^{A1}; A58-GAL4, UAS-PLC^{\delta}-PH-GFP, ppk-CD4-tdTomato^{10A}$

**Experimental genotype (Fig 2).** <u>*wild type*</u>: $w^{1118}$;; *A58-GAL4, UAS-PLC$^\delta$-PH-GFP, ppk-CD4-tdTomato$^{10A}$*

<u>*miR-14*</u>: $w^{1118}$; *miR-14$^{\Delta 1}$; A58-GAL4, UAS-PLC$^\delta$-PH-GFP, ppk-CD4-tdTomato$^{10A}$*

**Experimental genotypes (Fig 3).** (B-D) <u>*wild type*</u>: *trol$^{zcl1973}$*;; *ppk-CD4-tdTomato$^{10A}$*

<u>*miR-14*</u>: *trol$^{zcl1973}$; miR-14$^{\Delta 1}$; ppk-CD4-tdTomato$^{10A}$*

(E-F) <u>*wild type*</u>: $w^{1118}$; *ppk-CD4-tdGFP$^{1b}$; A58-GAL4, UAS-tdTomato / +*

<u>*miR-14*</u>: $w^{1118}$; *miR-14$^{\Delta 1}$, ppk-CD4-tdGFP$^{1b}$; A58-GAL4, UAS-tdTomato / +*

(G-K) <u>*wild type*</u>: $w^{1118}$; *ppk-CD4-tdGFP$^{1b}$, shg$^{mCherry}$*

<u>*miR-14*</u>: $w^{1118}$; *miR-14$^{\Delta 1}$, ppk-CD4-tdGFP$^{1b}$, shg$^{mCherry}$*

**Experimental genotypes (Fig 4).** (A-B) <u>*miR-14-GAL4*</u>: *Nrg$^{GFP}$; miR-14-GAL4 / +; UAS-tdTomato / +*

(C-E) <u>control sensor</u>: $w^{1118}$; *Tub-GFP / +; sr-GAL4, UAS-tdTomato / +*

<u>*miR-14*</u> <u>sensor</u>: $w^{1118}$; *Tub-GFP.miR-14 / +; sr-GAL4, UAS-tdTomato / +*

(F) <u>control</u>: *SOP-FLP, UAS-mCD8-GFP, 5-40-GAL4; FRT42D, elav-GAL80 x FRT42D*

<u>*drosha*</u>: *SOP-FLP, UAS-mCD8-GFP, 5-40-GAL4; FRT42D, elav-GAL80 x FRT42D, drosha$^{R662X}$*

<u>*miR-14*</u>: *SOP-FLP, UAS-mCD8-GFP, 5-40-GAL4; FRT42D, elav-GAL80 x FRT42D, miR-14$^{\Delta 1}$*

(G) $w^{1118}$; *ppk-CD4-tdGFP$^{1b}$ / UAS-mCherry.scramble.sponge; Act5C-GAL4 /+*

$w^{1118}$; *ppk-CD4-tdGFP$^{1b}$ / UAS-mCherry.miR-14.spongeV2; Act5C-GAL4 /+*

$w^{1118}$; *ppk-CD4-tdGFP$^{1b}$ / UAS-mCherry.scramble.sponge; A58-GAL4 / +*

$w^{1118}$; *ppk-CD4-tdGFP$^{1b}$ / UAS-mCherry.miR-14.spongeV2; A58-GAL4 /+*

$w^{1118}$, *5-40-GAL4; ppk-CD4-tdGFP$^{1b}$ / UAS-mCherry.scramble.sponge*

$w^{1118}$, *5-40-GAL4; ppk-CD4-tdGFP$^{1b}$ / UAS-mCherry.miR-14.spongeV2.sponge*

(H) $w^{1118}$; *miR-14$^{\Delta 1}$, ppk-CD4-tdGFP$^{1b}$ / miR-14$^{k10213}$; UAS-LUC-miR-14*

$w^{1118}$; *miR-14$^{\Delta 1}$, ppk-CD4-tdGFP$^{1b}$ / miR-14$^{k10213}$; UAS-LUC-miR-14 / Act5c-GAL4*

$w^{1118}$; *miR-14$^{\Delta 1}$, ppk-CD4-tdGFP$^{1b}$ / miR-14$^{k10213}$; UAS-LUC-miR-14 / A58-GAL4*

$w^{1118}$, *5-40-GAL4; miR-14$^{\Delta 1}$, ppk-CD4-tdGFP$^{1b}$ / miR-14$^{k10213}$; UAS-LUC-miR-14*

**Experimental genotypes (Fig 5).** (A-B) <u>*wild type*</u>: $w^{1118}$

<u>*Dcr1*</u>: $w^{1118}$;; Dcr1$^{mn29}$

<u>*miR-14$^{\Delta 1}$*</u>: $w^{1118}$; *miR-14$^{\Delta 1}$*

<u>*miR-14 $^{\Delta 1/k}$*</u>: $w^{1118}$; *miR-14$^{\Delta 1}$ / miR-14$^{k10213}$*

(C) <u>*wild type*</u>: $w^{1118}$

<u>*miR-14$^{\Delta 1}$*</u>: $w^{1118}$; *miR-14$^{\Delta 1}$*

<u>*nompC$^{1/3}$*</u>: *nompC$^1$, cn$^1$, bw$^1$ / nompC$^3$, cn$^1$, bw$^1$*

(D) <u>*wild type*</u>: $w^{1118}$

<u>*miR-14$^{\Delta 1}$*</u>: $w^{1118}$; *miR-14$^{\Delta 1}$*

<u>*nan$^{GAL4}$*</u>: $w^{1118}$; *nan$^{GAL4}$*

(E) <u>*wild type*</u>: $w^{1118}$

<u>*miR-14$^{\Delta 1}$*</u>: $w^{1118}$; *miR-14$^{\Delta 1}$*

(F) $w^{1118}$; *ppk-CD4-tdGFP$^{1b}$ / UAS-mCherry.scramble.sponge; Act5C-GAL4 /+*

$w^{1118}$; *ppk-CD4-tdGFP$^{1b}$ / UAS-mCherry.miR-14.spongeV2; Act5C-GAL4 /+*

$w^{1118}$; *ppk-CD4-tdGFP$^{1b}$ / UAS-mCherry.scramble.sponge; A58-GAL4 / +*

$w^{1118}$; *ppk-CD4-tdGFP$^{1b}$ / UAS-mCherry.miR-14.spongeV2; A58-GAL4 /+*

$w^{1118}$, *5-40-GAL4; ppk-CD4-tdGFP$^{1b}$ / UAS-mCherry.scramble.sponge*

$w^{1118}$, *5-40-GAL4; ppk-CD4-tdGFP$^{1b}$ / UAS-mCherry.miR-14.spongeV2.sponge*

(G) $w^{1118}$; *miR-14$^{\Delta 1}$, ppk-CD4-tdGFP$^{1b}$ / miR-14$^{k10213}$; Act5c-GAL4 / +*

$w^{1118}$; *miR-14$^{\Delta 1}$, ppk-CD4-tdGFP$^{1b}$ / miR-14$^{k10213}$; Act5c-GAL4 / UAS-LUC-miR-14*

$w^{1118}$; miR-14$^{\Delta 1}$, ppk-CD4-tdGFP$^{1b}$ / miR-14$^{k10213}$; Act5c-GAL4 / +

$w^{1118}$; miR-14$^{\Delta 1}$, ppk-CD4-tdGFP$^{1b}$ / miR-14$^{k10213}$; Act5c-GAL4 / UAS-LUC-miR-14

$w^{1118}$, 5-40-GAL4; miR-14$^{\Delta 1}$, ppk-CD4-tdGFP$^{1b}$ / miR-14$^{k10213}$; +

$w^{1118}$, 5-40-GAL4; miR-14$^{\Delta 1}$, ppk-CD4-tdGFP$^{1b}$ / miR-14$^{k10213}$; UAS-LUC-miR-14

(H-I) <u>wild type</u>: $w^{1118}$

<u>miR-14$^{\Delta 1}$</u>: $w^{1118}$; miR-14$^{\Delta 1}$

(I) <u>miR-14$^{\Delta 1}$</u>: $w^{1118}$; miR-14$^{\Delta 1}$

<u>miR-14$^{\Delta 1}$, TrpA1</u>: $w^{1118}$; miR-14$^{\Delta 1}$; TrpA1$^1$

<u>miR-14$^{\Delta 1}$, pain</u>: $w^{1118}$; miR-14$^{\Delta 1}$, pain$^1$

<u>miR-14$^{\Delta 1}$, Piezo</u>: $w^{1118}$; Piezo$^{KO}$, miR-14$^{\Delta 1}$

<u>miR-14$^{\Delta 1}$, ppk</u>: $w^{1118}$; ppk$^{ESB}$, miR-14$^{\Delta 1}$

**Experimental genotypes (Fig 6).** (A-G) <u>C4da-GAL4</u>: $w^{1118}$; shg$^{mCherry}$; ppk-GAL4 / +

<u>C4da>Integrins</u>: $w^{1118}$; shg$^{mCherry}$; ppk-GAL4 / UAS-mew, UAS-mys

<u>miR-14</u>: $w^{1118}$; miR-14$^{\Delta 1}$, shg$^{mCherry}$; ppk-GAL4 / +

<u>miR-14 + C4da>Integrins</u>: $w^{1118}$; miR-14$^{\Delta 1}$; ppk-GAL4 / UAS-mew, UAS-mys

**Experimental genotypes (Fig 7).** (A) <u>wild type</u>: $w^{1118}$

<u>miR-14$^{\Delta 1}$</u>: $w^{1118}$; miR-14$^{\Delta 1}$

(D-F) $w^{1118}$; miR-14$^{\Delta 1}$ / +

$w^{1118 / 67c23}$; miR-14$^{\Delta 1}$ / Inx2$^{G0118}$

$w^{1118}$ / ogre$^1$; miR-14$^{\Delta 1}$ / +

$w^{1118}$; miR-14$^{\Delta 1}$ / miR-14$^{\Delta 1}$

(G) $w^{1118}$; miR-14$^{\Delta 1}$ / +

$w^{1118}$ / ogre$^1$

$w^{1118 / 67c23}$; Inx2$^{G0118}$/ +

$w^{1118}$ / ogre$^1$; miR-14$^{\Delta 1}$ / +

$w^{1118 / 67c23}$; miR-14$^{\Delta 1}$ / Inx2$^{G0118}$

(H-Q) ogre$^{V5}$

ogre$^{V5}$; miR-14$^{\Delta 1}$, ppk-CD4-tdGFP$^{1b}$

(R-S) $w^{1118}$; miR-14$^{\Delta 1}$, ppk-CD4-tdGFP$^{1b}$; R38F11-GAL4, UAS-tdTomato / +

$w^{1118}$; miR-14$^{\Delta 1}$, ppk-CD4-tdGFP$^{1b}$; R38F11-GAL4, UAS-tdTomato / UAS-ogre

(T-U) <u>control</u>: $w^{1118}$; ppk-CD4-tdGFP$^{1b}$

<u>no UAS</u>: $w^{1118}$; miR-14$^{\Delta 1}$, ppk-CD4-tdGFP$^{1b}$; R38F11-GAL4, UAS-tdTomato / +

<u>Epi>ogre</u>: $w^{1118}$; miR-14$^{\Delta 1}$, ppk-CD4-tdGFP$^{1b}$; R38F11-GAL4, UAS-tdTomato / UAS-ogre

<u>Epi>inx2</u>: $w^{1118}$; miR-14$^{\Delta 1}$, ppk-CD4-tdGFP$^{1b}$; R38F11-GAL4, UAS-tdTomato / UAS-Inx2

(V) <u>control</u>: $w^{1118}$; en2.4-GAL4, UAS-RedStinger, ppk-CD4-tdGFP$^{1b}$/+; UAS-LUC-RNAi/+

<u>ogre RNAi</u>: $w^{1118}$; en2.4-GAL4, UAS-RedStinger, ppk-CD4-tdGFP$^{1b}$/+; UAS-ogreRNAi/+

<u>inx2 RNAi</u>: $w^{1118}$; en2.4-GAL4, UAS-RedStinger, ppk-CD4-tdGFP$^{1b}$/+; UAS-Inx2-RNAi/+

**Experimental genotypes (Fig 8).** (A-D) <u>Control</u>: $w^{1118}$;; ppk-LexA, AOP-GCaMP6s

<u>miR-14$^{\Delta 1}$</u>: $w^{1118}$; miR-14$^{\Delta 1}$; ppk-LexA, AOP-GCaMP6s

(E-F) <u>Control</u>: $w^{1118}$;; ppk-GAL4, UAS-Gerry / +

<u>miR-14$^{\Delta 1}$</u>: $w^{1118}$; miR-14$^{\Delta 1}$; ppk-GAL4, UAS-Gerry / +

<u>miR-14$^{\Delta 1}$ + Integrins</u>: $w^{1118}$; miR-14$^{\Delta 1}$; ppk-GAL4, UAS-Gerry / UAS-mew, UAS-mys

(G) $w^{1118}$; Mhc::Cherry/+; ppk-CD4-tdTomato$^{10A}$ / Ilk$^{ZCL3111}$

(H-K) <u>Control</u>: $w^{1118}$ ;; Nrx-IV$^{GFP}$, ppk-CD4-tdTomato$^{10A}$

<u>Tig$^{A1/X}$</u>: $w^{1118}$; Tig$^{A1}$ / Tig$^X$; Nrx-IV$^{GFP}$, ppk-CD4-tdTomato$^{10A}$

(L) <u>Control</u>: $w^{1118}$ ; Tig$^{A1}$ / +

<u>Tig$^{A1/X}$</u>: $w^{1118}$ ; Tig$^{A1}$ / Tig$^X$

(M) <u>Control</u>: $w^{1118}$ ; Tig$^{A1}$ / +; ppk-GAL4, UAS-Gerry

$Tig^{A1/X}$: $w^{1118}$; $Tig^{A1}$ / $Tig^X$; ppk-GAL4, UAS-Gerry

## Supporting information

**S1 Fig. Genetic screen for mutations that alter dendrite positioning over epidermal cells.**
(A-D). Maximum intensity projections show C4da dendrites (*ppk-CD4-tdTomato*) of pre-EMS control (A, C) and $Dcr1^{dg29}$ mutant larvae (B, D) at 120 h AEL that have been pseudocolored to show areas of dendrite alignment along epidermal junctions ($Nrx-IV^{GFP}$). Insets (A' and B') show high magnification views of dendrite arbors, highlighting corresponding regions from control and $Dcr1^{dg29}$ mutant larvae that contain junction-aligned dendrites. Arrows mark dendrite crossing events involving junction-aligned dendrites and dashed lines mark aligned dendrites that are bundled. (C, D) Maximum intensity projections show relative positions of C4da neurons and apodemes (asterisks). In both wild-type control (C) and (D) $Dcr1^{dg29}$ mutant larvae, dendrites intercalate between and wrap around apodemes, but rarely innervate below apodemes. (E-H) Morphometric analysis of $Dcr1^{dg29}$ C4da dendrite positioning defects. (E) Plot depicts the proportion of C4da dendrite arbors, excluding regions covering apodemes, that align along epidermal junctions. (F-H) Junction-aligned dendrites are frequently involved in homotypic dendrite crossing events. Plots depict (F) the frequency of homotypic dendrite crossing events within a C4da dendrite arbor (crossing number normalized to mm total dendrite length), (G) the proportion of junctional alignment sites within a C4da dendrite arbor that contain dendrite-dendrite crossing events, and (H) the proportion of homotypic dendrite crossing events within a C4da dendrite arbor that occur at sites of epidermal dendrite alignment. (I) Screen for miRNAs that control epidermal dendrite alignment. Deficiency alleles cover 130 miRNA genes, accounting for >99% of somatically expressed miRNAs [108]. (J-L) Representative images of C4da neurons from segment A3 at 96 h AEL in wild-type control (J), $Dcr-1^{dg29}$ (K), and $miR-14^{Δ1}$ mutant larvae (L). (M-Q) Morphometric analysis of C4da dendrites in $Dcr-1^{dg29}$ and $miR-14^{Δ1}$ mutant larvae. Mutations in $Dcr-1^{dg29}$ and $miR-14^{Δ1}$ have no significant effect on total dendrite length (M), dendrite branch points (N) or terminal dendrite number (O), but exhibit progressive deficits in dendrite coverage (P) and a progressive increase in dendrite-dendrite crossing events (Q). *$P<0.05$ compared to pre-EMS control; unpaired t-test with Welch's correction (E-G), Mann Whitney test (H), Kruskal-Wallis test with post-hoc Dunn's test (M-O, Q), or ANOVA with post-hoc Tukey's test (P). Experimental genotypes: (A-H) *wild type*: $w^{1118}$;; $Nrx-IV^{GFP}$, ppk-CD4-tdTomato$^{10A}$, *Dcr1*: $w^{1118}$;; $Nrx-IV^{GFP}$, ppk-CD4-tdTomato$^{10A}$, $Dcr1^{mn29}$, (J-Q) *wild type*: $w^{1118}$;; ppk-CD4-tdTomato$^{10A}$, *Dcr1*: $w^{1118}$;; $Dcr1^{mn29}$, ppk-CD4-tdTomato$^{10A}$, *miR-14*: $w^{1118}$; $miR-14^{Δ1}$; ppk-CD4-tdTomato$^{10A}$.
(TIFF)

**S2 Fig. Epidermal junction alignment and epidermal ensheathment of dendrites are distinct phenomena.** (A) Apodeme innervation is unaffected in miR-14 mutants. Maximum intensity projections show relative positions of C4da neurons and apodemes (asterisks) in wild-type control and *miR-14* mutant larvae. In both backgrounds, dendrites intercalate between and wrap around apodemes (pseudocolored magenta), but rarely innervate below apodemes. (B) Plot depicts the total length of C4da dendrites aligned to non-apodeme epidermal junctions at the indicated developmental timepoints. *$P<0.05$, Kruskal-Wallis test with post-hoc Dunn's test. (C-D) Distribution of epidermal junctional alignment in C4da dendrite arbors. (C) Maximum intensity projection of representative $miR-14^{Δ1}$ mutant C4da neuron in which dendrites are pseudocolored cyan to indicate sites of epidermal junctional alignment, green to mark aligned terminal dendrites, and magenta to indicate aligned stretches that do not involve terminal dendrites. Dendritic alignment along epidermal junctions was identified

by dendritic CD4-tdTomato (*ppk-CD4-tdTomato*) co-localization with the PIP2 marker PLC$^\delta$-PH-GFP. (D) Plot depicts the fraction of epidermal junction-aligned dendrites that involve terminal (green) and non-terminal (magenta) dendrites. *P<0.05, unpaired t-test with Welch's correction. (E-H) *miR-14* does not affect epidermal distribution of C1da or C3da dendrites. (E-F) Maximum intensity projections of wild-type control and *miR-14* mutant larvae expressing *shg$^{RFP}$*, which labels epidermal cell-cell junctions, and membrane-targeted *CD4-tdGFP* expressed in C1da neurons (E) or C3da neurons (F). Plots depict (G) total dendrite length and (H) proportion of dendrites that are aligned along epidermal junctions for the indicated genotype-cell type combinations. Comparing values from control and *miR-14* mutant larvae using a Mann Whitney test revealed no significant differences. (I) Epidermal dendrite ensheathment and epidermal junctional dendrite alignment are regulated by distinct genetic pathways. Maximum intensity projections show representative C4da neurons from *ban$^{\Delta 1}$* mutant and *miR-14$^{\Delta 1}$; ban$^{\Delta 1}$* double mutant larvae. C4da neurons from the left side of the larvae are in the center of the field of view, the dorsal midline is indicated with a white hatched line, and dendrites of contralateral neurons are pseudocolored yellow to highlight the *ban$^{\Delta 1}$* mutant dendrite coverage defects. Both *ban$^{\Delta 1}$* mutant and *miR-14$^{\Delta 1}$; ban$^{\Delta 1}$* double mutant larvae exhibit intermingling of adjacent dendrite arbors. In addition, portions of the dendrite arbor that were apically shifted >1.5 mm from neighboring branches, which correspond to junction-aligned dendrites, are pseudocolored magenta. Unlike *ban$^{\Delta 1}$* mutants, *miR-14$^{\Delta 1}$; ban$^{\Delta 1}$* double mutants exhibit extensive apical dendrite displacement. (J) Plot depicts the frequency of C4da neuron dendrite-dendrite crossing events in *ban* mutant and *miR-14; ban* double mutant larvae. *P<0.05, Mann Whitney test. Experimental genotypes: (A-B) <u>wild type</u>: *w$^{1118}$;; Nrx-IV$^{GFP}$, ppk-CD4-tdTomato$^{10A}$*, <u>miR-14</u>: *w$^{1118}$;; miR-14$^{\Delta 1}$, Nrx-IV$^{GFP}$, ppk-CD4-tdTomato$^{10A}$*, (C-D) <u>miR-14</u>: *w$^{1118}$; miR-14$^{\Delta 1}$; A58-GAL4, UAS-PLC$^\delta$-PH-GFP, ppk-CD4-tdTomato$^{10A}$*, (E-H) <u>C1da wild type</u>: *w$^{1118}$; 98b-GAL4, UAS-CD4-tdGFP$^{8M2}$, shg$^{mCherry}$*, <u>C1da miR-14</u>: *w$^{1118}$; 98b-GAL4, UAS-CD4-tdGFP$^{8M2}$, miR-14$^{\Delta 1}$, shg$^{mCherry}$*, <u>C3da wild type</u>: *w$^{1118}$; nompC-GAL4$^P$, UAS-CD4-tdGFP$^{8M2}$, shg$^{mCherry}$* <u>C3da miR-14</u>: *w$^{1118}$; nompC-GAL4$^P$, UAS-CD4-tdGFP$^{8M2}$, miR-14$^{\Delta 1}$, shg$^{mCherry}$*, (I-J) <u>ban$^{\Delta 1}$</u>: *w$^{1118}$;; ppk-CD4-tdGFP$^{1b}$; ban$^{\Delta 1}$*, <u>miR-14$^{\Delta 1}$; ban$^{\Delta 1}$</u>: *w$^{1118}$; miR-14$^{\Delta 1}$, ppk-CD4-tdGFP$^{1b}$; ban$^{\Delta 1}$*.
(TIFF)

**S3 Fig. Time-lapse imaging of new dendrite branch alignment relative epidermal junctions.** C4da neurons (*ppk-CD4-tdTomato*) were imaged over a 24 h time-lapse (96–120 h AEL) and the orientation of branch growth in relation to epidermal junctions (*A58-GAL4, UAS-PLC$^\delta$-PH-GFP*) was monitored for each new dendrite branch. Composite montages from representative for (A) wild-type control and (B) *miR-14* mutant larvae show new dendrite branches pseudocolored according to following orientations: growth towards (green), away from (magenta) or aligned along epidermal junctions (cyan). Raw images for each genotype / time point combination are shown below composites. Experimental genotypes: <u>wild type</u>: *w$^{1118}$;; A58-GAL4, UAS-PLC$^\delta$-PH-GFP, ppk-CD4-tdTomato$^{10A}$*, <u>miR-14</u>: *w$^{1118}$; miR-14$^{\Delta 1}$; A58-GAL4, UAS-PLC$^\delta$-PH-GFP, ppk-CD4-tdTomato$^{10A}$*.
(TIFF)

**S4 Fig. Using *miR-14-GAL4* to express *miR-14* rescues *miR-14* mutant dendrite positioning defects.** Composite images show morphology (white) and epidermal junctional alignment (magenta) of C4da dendrites in *miR-14* mutant larvae that additionally express *miR-14-GAL4* without (A) or with (B) *UAS-miR-14*. (C) *UAS-miR-14* expression driven by *miR-14-GAL4* rescues the *miR-14* mutant junctional dendrite alignment defect. *P<0.05, unpaired t-test with Welch's correction. (D-F) Related to Fig 4F. Maximum intensity projections show representative images of C4da neuron MARCM clones of the indicated genotypes. (G-L) Related to

Fig 4G. Effects of miRNA sponge expression. C4da neurons from larvae expressing *miR-14* sponge (G-I) or control sponge (J-L) with the indicated *GAL4* driver. (M-O) Related to Fig 4H. *miR-14* rescue assays. Maximum intensity projections show representative images of C4da neurons in *miR-14* mutant larvae expressing *UAS-miR-14* with the indicated *GAL4* drivers. Experimental genotypes: (A-C) $w^{1118}$; *miR-14-GAL4, miR-14$^{Δ1}$ / miR-14$^{k10213}$, $w^{1118}$; miR-14-GAL4, miR-14$^{Δ1}$ / miR-14$^{k10213}$; UAS-LUC-miR-14*, (D-F) Genotypes listed in Fig 4F, (G-L) Genotypes listed in Fig 4G, (M-P) Genotypes listed in Fig 4H.
(TIFF)

**S5 Fig. *miR-14* does not broadly repress apodeme cell fate.** (A) Plots depict mRNA expression levels of apodeme marker genes in RNA-seq libraries generated from dissociated, manually picked apodemes (orange) and epidermal cells (teal). See also S1 Table for a complete list of genes differentially expressed between apodemes and epidermal cells. (B) *miR-14* does not regulate expression of apodeme genes in epidermal cells. Venn diagram depicts overlap of genes differentially expressed between control and *miR-14* mutant epidermal cells (yellow circle) and genes differentially expressed between apodemes and epidermal cells (blue circle). The intersection between the two datasets contains only five genes, none of which have known function in apodeme development or cell fate. (C-E) Ectopic expression of *miR-14* in apodemes promotes denrite invasion into apodeme domains. (C-D) Representative maximum projection images from confocal stacks depict the distribution of C4da dendrites (*ppk-CD4-tdGFP*) over apodemes (*sr-GAL4, UAS-DsRed*) at segment boundaries in control larvae (C) or larvae overexpressing *miR-14* selectively in apodemes (D). Apodeme boundaries are marked by hatched lines. (E) Violin plot depicts the density of dendrite invasion on the basal surface of apodemes. Each point represents a measurement of dendrite coverage from an individual apodeme. P = 1.85 x $10^{-11}$, unpaired T-test with Welch's correction. Experimental genotypes: (A-B) Epidermal cells: $w^{1118}$; *R38F11-GAL4 / UAS-2x-EGFP, miR-14* epidermal cells: $w^{1118}$; *miR-14$^{Δ1}$ / miR-14$^{k10213}$; R38F11-GAL4 / UAS-2x-EGFP*, Apodemes: $w^{1118}$; *UAS-2x-EGFP/+; sr-GAL4 / +*, (C-E) *sr>DsRed*: *ppk-CD4-tdGFP / +; sr-GAL4 / UAS-DsRed, sr>DsRed-miR-14*: *ppk-CD4-tdGFP / +; sr-GAL4 / UAS-DsRed-miR-14*.
(TIFF)

**S6 Fig. Supplement to Fig 5.** (A) C4da neuron activity is required for *miR-14* mutant hypersensitivity to mechanical stimuli. Expressing *UAS-Kir2.1* in nociceptive C4da neurons suppresses *miR-14* mutant mechanical hypersensitivity. Plots depict the proportion rolling (left) and mean roll number (right) of larvae of the indicated genotype to 80 mN von Frey stimulation. *P<0.05, Fisher's exact test with a BH correction (proportion rolling) or Kruskal-Wallis test followed by Dunn's multiple comparisons test (roll number). (B) *Dcr1* mutants and an additional *miR-14* allelic combination exhibit enhanced nocifensive behavior responses. Plots depict proportion of larvae that exhibit nocifensive rolling and mean number of nocifensive rolls in response to von Frey fiber stimulation of the indicated intensities. *P<0.05, Fisher's exact test with a BH correction (proportion rolling) or Kruskal-Wallis test followed by Dunn's multiple comparisons test (roll number). (C) *UAS-miR-14* expression driven by *miR-14-GAL4* rescues the *miR-14* mutant nociception sensitization phenotype. *P<0.05, Fisher's exact test. (D) *miR-14* is dispensable for UV-induced thermal allodynia. Plots depict latency values (seconds) to the first nociceptive roll in response to 38°C stimuli for control and *miR-14* mutant larvae 24 h following mock treatment or UV irradiation. Chi-square test revealed no significant difference between the two genotypes. (E) Epidermal expression of the TNF ligand Eiger is dispensable for nociceptive sensitization in *miR-14* mutant larvae. Plot depicts nociceptive rolling responses to 25 mN stimulus of *miR-14* mutant larvae expressing *Luciferase-RNAi* or *eiger-RNAi* in epidermal cells. NS, no signicant difference, Fisher's exact test. (F) *Piezo* is

largely dispensable for nociceptive rolling responses induced by a 25mN stimulus. Plots depict nocifensive rolling responses to 25 mN von Frey fiber stimulation of control or channel mutant larvae. *P < 0.05, Kruskal-Wallis test followed by a Dunn's multiple comparisons test. (G-I) Mutation of *Piezo* suppresses *miR-14* nociceptive sensitization but not epidermal dendrite intercalation defects. Maximum intensity projections show representative images of C4da neurons from (G) *miR-14* mutant and *miR-14, Piezo* double mutant larvae. *Piezo* mutation has no significant effect on (H) total dendrite length or (I) dendrite-dendrite crossing frequency in *miR-14* mutant larvae. Not significant, unpaired t-test with Welch's correction. Experimental genotypes: (A) $w^{1118}$, $w^{1118}$;; ppk-GAL4 / UAS-TnT $w^{1118}$; $miR-14^{\Delta1}$ / miR-$14^{k10213}$; ppk-GAL4 / UAS-TnT, (B) $w^{1118}$ $w^{1118}$;; $Dcr1^{mn29}$, $w^{1118}$; $miR-14^{\Delta1}$ / $miR-14^{k10213}$, (C) $w^{1118}$; miR-14-GAL4, $miR-14^{\Delta1}$ / $miR-14^{k10213}$, $w^{1118}$; miR-14-GAL4, $miR-14^{\Delta1}$ / $miR-14^{k10213}$; UAS-LUC-miR-14, (D) *wild type*: $w^{1118}$, $miR-14^{\Delta1}$: $w^{1118}$; $miR-14^{\Delta1}$, (E) $w^{1118}$; $miR-14^{\Delta1}$ / miR-14k10213; A58-GAL4, $w^{1118}$; $miR-14^{\Delta1}$ / miR-14k10213; A58-GAL4 / UAS-eiger$^{IR}$, (F) *control*: $w^{1118}$; $miR-14^{\Delta1}$, *TrpA1*: $w^{1118}$; $TrpA1^1$, *pain*: $w^{1118}$; $pain^1$, *Piezo*: $w^{1118}$; $Piezo^{KO}$, *ppk*: $w^{1118}$; $ppk^{ESB}$, (G-I) *miR-14*: $miR-14^{\Delta1}$, ppk-CD4-tdGFP, *miR-14, Piezo*: $miR-14^{\Delta1}$, $Piezo^{\Delta}$, ppk-CD4-tdGFP.
(TIFF)

**S7 Fig. *mda1* mutation enhances of *miR-14* mutant dendrite intercalation and nociception phenotypes.** Representative composite images of C4da dendrites pseudocolored cyan at sites of epidermal junctional alignment are shown for (A) $miR-14^{\Delta1}$ mutant and (B) $miR-14^{\Delta1}$, $mda1^{246}$ double mutant larvae. Sites of dendrite alignment to epidermal junction were identified as sites of colocalization between GFP (*ppk-CD4-tdGFP*) and anti-cora immunoreactivity. (C) Plot depicts the proportion of dendrite arbors aligned along epidermal junctions in larvae of the indicated genotypes. *P<0.05, unpaired t-test with Welch's correction. (D) Plot depicts nociceptive rolling responses of larvae of the indicated genotypes to 25 mN von Frey stimulation. *P<0.05, Fisher's exact test. Experimental genotypes: (A-D) $w^{1118}$; $miR-14^{\Delta1}$, ppk-CD4-tdGFP$^{1b}$, $w^{1118}$; $miR-14^{\Delta1}$, $mda1^{246}$, ppk-CD4-tdGFP$^{1b}$.
(TIFF)

**S8 Fig. *miR-14* regulates epidermal Inx distribution.** (A) Epidermal expression of *Inx* genes. Plot depicts mean and standard deviaion for mRNA levels (normalized counts) for the indicated Innexin genes from RNA-seq analysis of epidermal cells. Points indicate expression values from independent biological replicates. (B) Morphometric analysis of C4da dendrites in larvae heterozygous for $miR-14^{\Delta1}$ and loss-of-function mutations in the indicated epidermal junction genes showing the mean number of dendrite-dendrite crossing events per neuron. *P<0.05 compared to *miR-14* heterozygous controls, ANOVA followed by post-hoc Dunnett's test. (C) Expression and distribution of AJ and SJ markers in control and *miR-14* mutant larvae. Representative images show expression of the AJ markers armadillo and discs large (C, D), and the septate junction markers coracle (E, F) and neuroglian (G, H) in individual epidermal cells of control or *miR-14* mutant larvae expressing *ppk-CD4-tdGFP* to label C4da dendrites. *miR-14* mutation did not cause substantial alterations in level or distribution of these markers. (D-E) *miR-14* regulates epidermal Inx2 distribution. Maximum intensity projections show distribution of Inx2 in the epidermis of wild type control (D) and *miR-14* mutant larvae (E). As with ogre immunoreactivity (Fig 7), *miR-14* mutation caused irregularities in the belt of Inx2 immunoreactivity including frequent discontinuities (arrows). (F-I) GJ proteins are differentially expressed at apodemes and other epidermal cells. Maximum intensity projections (top) show C4da dendrites in green and the GJ proteins ogre (F) and Inx2 (G) in magenta, and lookup tables depict Inx intensity. Hatched lines outline apodeme boundaries, and cell-cell interfaces between apodemes (Apo) and epidermal cells (Epi) are indicated with arrows. Plots

depict (H) ogre and Inx2 intensity and (I) the cross-sectional width of ogre and Inx2 immuno-reactivity at cell-cell interfaces of apodemes and other epidermal cells. *$P<0.05$, Kruskal-Wallis test followed by Dunn's multiple comparisons test. Experimental genotypes: (A) $w^{1118}$; miR-$14^{\Delta 1}$ / +, $w^{1118}$ / +; miR-$14^{\Delta 1}$ / $cn^1$, $shg^2$, $bw^1$, $sp^1$, $y^1$, $w^{1118}$, $dlg1^A$, FRT19A / $w^{1118}$; miR-$14^{\Delta 1}$ / +, $w^{1118}$ / $Nrg^{14}$; miR-$14^{\Delta 1}$ / +, $y^1$, $w^{1118}$ / $w^{1118}$; miR-$14^{\Delta 1}$ / $kune^{C309}$, $w^{1118}$; miR-$14^{\Delta 1}$ / FRT43D, $cora^5$, $w^{1118}$ / $ogre^1$ ; miR-$14^{\Delta 1}$ / +, $w^{1118 / 67c23}$ ; miR-$14^{\Delta 1}$ / $Inx2^{G0118}$, (B) $w^{1118}$ ;; R38F11-GAL4 / UAS-nls-GFP, (C) $w^{1118}$; ppk-CD4-tdGFP$^{1b}$, $w^{1118}$; miR-$14^{\Delta 1}$, ppk-CD4-tdGFP$^{1b}$, Nrg specimens additionally contained Nrg$^{GFP}$/+, (D) Inx2$^{V5}$; ppk-CD4-tdGFP$^{1b}$, (E) Inx2$^{V5}$; miR-$14^{\Delta 1}$, ppk-CD4-tdGFP$^{1b}$, (F-I) ogre$^{V5}$; ppk-CD4-tdGFP$^{1b}$, Inx2$^{V5}$; ppk-CD4-tdGFP$^{1b}$
(TIFF)

**S9 Fig. Epidermal *Inx* knockdown.** Representative images depict effects of *UAS-ogre-RNAi* (A-B) or *UAS-Inx2-RNAi* (C-D) expression on Inx protein levels. (A, C) Maximum intensity projections show anti-GFP immunoreactivity to label the *en-GAL4* expression domain and (B, D) anti-V5 immunoreactivity to label V5-tagged endogenous ogre (B) or Inx2 (D), (B', D') Inx protein levels pseudocolored according to a lookup table. White hatched lines demarcate boundaries of *en-GAL4* expression domains. (E) Plot depicts ogre and Inx2 intensity at epidermal junctions outside (GAL4-) or inside (GAL4+) the *en-GAL4, UAS-RNAi* expression domain. *$P<0.05$, ANOVA with post-hoc Sidak's test. (F-H) Double labeling of C4da dendrites (*ppk-CD4-tdGFP*) and epidermal cells (*en-GAL4, UAS-NLS-GFP*) additionally expressing (F) *UAS-LUC-RNAi* (control), (G) *UAS-Inx2-RNAi*, or (H) *UAS-ogre-RNAi*. Traces show dendrite arborization and epidermal junction-aligned dendrites (pseudocolored green) within the *en-GAL4* expression domain. Experimental genotypes: (A-E) <u>ogre RNAi</u>: ogre$^{V5}$; en2.4-GAL4, UAS-RedStinger, ppk-CD4-tdGFP$^{1b}$/+; UAS-ogreRNAi/+, <u>Inx2 RNAi</u>: Inx2$^{V5}$; en2.4-GAL4, UAS-RedStinger, ppk-CD4-tdGFP$^{1b}$/+; UAS-Inx2-RNAi/+, (F-H) <u>LUC RNAi</u>: $w^{1118}$; en2.4-GAL4, UAS-RedStinger, ppk-CD4-tdGFP$^{1b}$/+; UAS-LUC-RNAi/+, <u>ogre RNAi</u>: $w^{1118}$; en2.4-GAL4, UAS-RedStinger, ppk-CD4-tdGFP$^{1b}$/+; UAS-ogreRNAi/+, <u>Inx2 RNAi</u>: $w^{1118}$; en2.4-GAL4, UAS-RedStinger, ppk-CD4-tdGFP$^{1b}$/+; UAS-Inx2-RNAi/+
(TIFF)

**S1 Table. Differential expression analysis of RNA-seq samples.** Expression levels and log2-foldchange values are shown for all genes that were significantly differentially expressed (adjusted p value < 0.05) for the following sample comparisons: control vs. *miR-14* mutant epidermal cells; apodemes vs. epidermal cells (both from wild-type control larvae).
(XLS)

**S2 Table. Alleles used in this study.** Each allele is listed according to its use in this study along with the corresponding RRID identifier.
(PDF)

**S1 Data. The raw data presented in this study are tabulated according to the corresponding figure panel from the manuscript.** Details of data analysis including tests of normality, statistical analysis, and post-hoc tests are provided.
(XLSX)

## Acknowledgments

We thank Andrea Brand and Stefanie Schirmeier for sharing *ogre* and *Inx2* alleles; Stephen Cohen for sharing *miR-14* sensor flies; Eric Lai for sharing *drosha* mutant flies; David van Vactor for sharing miRNA sponge transgenic flies; Marvin Nayan for assistance with genetic analysis of *Dcr1* mutants; Keegan McElligot for assistance with nociceptive behavior assays; Amy

Platenkamp for assistance with image analysis; Peter Soba for guidance and support in behavior assays; and the Parrish lab for helpful discussions.

## Author Contributions

**Conceptualization:** Kory P. Luedke, Jay Z. Parrish.

**Data curation:** Kory P. Luedke, Chang Yin, Jay Z. Parrish.

**Formal analysis:** Kory P. Luedke, Jiro Yoshino, Chang Yin, Nan Jiang, Jessica M. Huang, Kevin Huynh, Jay Z. Parrish.

**Funding acquisition:** Kory P. Luedke, Jay Z. Parrish.

**Investigation:** Kory P. Luedke, Jiro Yoshino, Nan Jiang, Jay Z. Parrish.

**Project administration:** Jay Z. Parrish.

**Resources:** Jay Z. Parrish.

**Supervision:** Jay Z. Parrish.

**Visualization:** Jiro Yoshino, Chang Yin, Nan Jiang.

**Writing – original draft:** Jay Z. Parrish.

**Writing – review & editing:** Kory P. Luedke, Jiro Yoshino, Chang Yin, Jessica M. Huang.

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
