## [Decision Letter · Decision Letter 0]

24 Oct 2023

Dear Dr Parrish,

Thank you very much for submitting your Research Article entitled 'Dendrite intercalation between epidermal cells tunes nociceptor sensitivity to mechanical stimuli in Drosophila larvae' to PLOS Genetics.

The manuscript was fully evaluated at the editorial level and by independent peer reviewers. The reviewers appreciated the attention to an important topic but identified some concerns that we ask you address in a revised manuscript.

We therefore ask you to modify the manuscript according to the review recommendations. Your revisions should address the specific points made by each reviewer.

Yours sincerely,

Fengwei Yu

Academic Editor

PLOS Genetics

Gregory P. Copenhaver

Editor-in-Chief

PLOS Genetics

Reviewer's Responses to Questions

**Comments to the Authors:**

Reviewer #1: In this manuscript, Luedke et al used c4 da dendrites as a model to study the junctional dendrite alignment. They discovered that there are more aligned junctional dendrite in Dicer1, and miR-14 mutant. The junctional dendrite alignment increased at 108hAEL and only happened in c4da neuron in miR-14 mutant. Junctional aligned dendrites in miR-14 mutant showed more outgrowth than retraction. Furthermore, they found out that the junctional dendrite alignment is different from dendrite ensheathment.

They also found out that miR-14 is expressed in epithelial cells excluding the apodemes to regulate dendrite alignment. Gap junction protein, Orge and Inx2 are reduced in miR-14 mutant and are likely targets regulated by miR-14, although likely indirectly. Gap junction protein reduction increases dendrite accessibility to epidermal intercellular space and/or junction in miR-14 mutant. However, gap junction proteins are not the target of miR-14.

Moreover, they found out that miR-14 mutant larvae showed higher frequency of nocifensive responses to stimulus. They concluded that epidermal dendrite intercalation would affect nociceptor sensitivity to mechanical stimuli.

This manuscript reports several interesting and novel findings that would benefit the community in the understanding of dendrite arborization and functional linkage. I suggest to publish the manuscript with proper modification, mainly for the ease of reading.

I have a few questions and suggestions:

1. In the manuscript, the authors only studied Ogre and Inx2 to relate miR14 function. From target scan database in mirbase, Inx3 is predicted as the direct target of miR-14. Have the authors checked whether the expression level of Inx3 is increased in miR-14 mutant? How about the genetic interaction with miR-14? If the inx3 is upregulated in miR-14 mutants, the increased levels could also interfere gap junction integrity and lead to dendrite alignment increase.

2. Fig. 1 and legend need revision:

(i) Fig 1A, histoblasts (indicated in Fig 1B) at left bottom are masked!

(ii) Fig 1E’ shows aligned dendrite segments that are supposed larger than 5 um (described in Image analysis). Some small ones are obviously smaller than 5um!

(iii) Legend does not match Figures, e.g. (H) shows aligned length is described as ensheathed, and also (I-K) are not either described or mis-described.

(iv) in Fig1S2C, the aligned non-terminals image (right panel) is not properly presented, as the terminal portions are cropped. This is misleading to readers when viewing the image as the magenta colored portions appear to be terminal!

3. Are there contributions to dendrite intercalation from epidermal attachment to muscle cells?

4. Fig 3E, the cytosolic RFP appears to be very punctated?

5. The conclusion in lines 369-370 is not well supported. The effect in suppressing miR14 mutants in both dendrite alignment to cell junctions and nociefensive rolling responses could be also due to dendrite attachment to basement membranes and/or enshealthment. (Since Integrins overexpression had no effect on dendrite intercalation in wild type background (Fig 6E), but suppressed nociefensive rolling responses (80mN in Fig 6F) in wild type larvae. I suggest the conclusion should be modified, and the figure can be moved to Supplement.

6. The inclusion of mda mutants does not provide much support as the information provided is limited. I suggest to skip the data in this manuscript.

7. The study of gap junction proteins shows transheterozygous genetic interaction, regulation of protein levels, and the suppression of miR14 mutant by overexpression indicate a major functional regulation of miR-14, and the involvement of gap junction protein in dendrite intercalation. These data are important, and I wonder why these data are not shown as main figures.

8. Finally, may I suggest a reorganization of Fig 1. The initial introduction of dendrite intercalation around apodemes is quite nice, but most of phenotypic analysis of mir-14 mutant in this manuscript is about the dendrite orientation defects in other epidermal cells. Also, mir-14 mutants present multiple defects in addition to increased dendrite intercalation. It would be nice the manuscript starting with genetic screens for Dcr1 and miR14. Then the intercalation phenotype could be introduced for further study and suggest they are similar to dendrites near apodemes.

Reviewer #2: The Drosophila larva detects nociceptive stimuli in the environment using class IV da neurons that have free dendrite endings. Although a lot has been known about the sensory functions and dendrite organization of these neurons on the body wall epidermis, how dendrite-epidermal cell spatial distribution and neuronal sensitivity are linked was previously not very well understood. In this manuscript by Luedke at al., the authors identified an interesting mechanism by which the intercalation of sensory dendrites in the epidermal cell lateral junction is regulated by miR-14. They found that miR-14 is expressed in higher levels in regular epidermal cells but is down regulated in epodemes, so that junction intercalation of dendrites is only present at epodemes. They further found that miR-14 does so by regulating gap junction proteins, including Ogre and Inx2, expressed by epidermal cells. Lastly, they found that junction intercalation of dendrites sensitizes neurons in ways that are independent of other known pathways of sensitization.

Overall, this is a very nice study presenting novel and interesting mechanisms of dendrite/epidermis interaction and neuronal function. The investigations are multilayered and are complementary with one another. The data presented are compressive and of high quality. The evidence for their main claims is generally compelling. I support acceptance of this paper after minor revisions through additional control experiments and text editing.

Major concerns:

1. In Figure 5, the authors showed the effects of removing various channels in the miR-14Δ1 background on mechanonociceptive response of larvae. These channel mutants should be examined in the wildtype background as well to serve as controls.

2. Does integrin overexpression in miR-14 mutant cause dendrite reduction (Figure 6D)? If so, the reduced sensitivity could also be due to dendrite reduction.

3. The authors claim that miR-14 regulates the expression and distribution of Ogre and Inx2, however, the only results presented are representative images in Figure 7L-O. To make such claims, quantitative analyses are necessary.

Minor concerns:

1. Is there anything else known about mda-1? Which gene does it encode? What kind of molecule is it? The information should be included if available.

2. The authors could consider including a little more context about the RNA-seq experiment. Currently it is only very briefly mentioned.

3. Figure 8: Are groups mislabeled or bars mis-color-coded in 8M? How is apodemes labeled in K-L’? Does Tig LOF cause changes in other parts of the neuron?

4. miR-14 is known to suppress cell death and to regulate fat metabolism and insulin production. The authors may want to discuss other possible ways in which miR-14 indirectly affects neuronal activity.

Reviewer #3: Attachment

**Have all data underlying the figures and results presented in the manuscript been provided?**

Reviewer #1: Yes

Reviewer #2: Yes

Reviewer #3: Yes

PLOS authors have the option to publish the peer review history of their article (what does this mean?). If published, this will include your full peer review and any attached files.

Reviewer #1: No

Reviewer #2: No

Reviewer #3: No

---

## [Decision Letter · Decision Letter 1]

22 Jan 2024

Dear Dr Parrish,

Thank you very much for submitting your Research Article entitled 'Dendrite intercalation between epidermal cells tunes nociceptor sensitivity to mechanical stimuli in Drosophila larvae' to PLOS Genetics.

The manuscript was fully evaluated at the editorial level and by independent peer reviewers. The reviewers appreciated the attention to an important topic but identified some concerns that we ask you address in a revised manuscript.

We therefore ask you to modify the manuscript according to the review recommendations. Your revisions should address the specific points made by each reviewer.

Yours sincerely,

Fengwei Yu

Academic Editor

PLOS Genetics

Gregory P. Copenhaver

Editor-in-Chief

PLOS Genetics

Reviewer's Responses to Questions

**Comments to the Authors:**

Reviewer #1: A minor correction is suggested: Line 364 (Fig 5E, 5F) should be (Fig. 5F, 5G) and Line 372 (Fig 5G) should be (Fig 5E) to cope with the figure arrangements.

Reviewer #2: The authors have successfully addressed my concerns. Congratulations on such a nice study!

Reviewer #3: The manuscript of Luedke et al has been greatly improved during the revision, a minor weakness remains in the treatment of the distinction of nociceptor dendrite innervation of apodeme/tendon cells relative to epidermal cell innervation. Recommend that this be revised further.

Minor Points

1.) The following statement by the authors is not consistent with published literature on the formation of the muscle tendon junction nor with what is known about the development of the dendritic arbors of the nociceptors:

“Whether these distinct orientations reflect physical occlusion of apodeme territory by muscles, the presence of attractive cues at apodeme junctional domains, and/or the repulsive cues at other epidermal cell-cell interfaces is currently unknown”.

The muscle tendon junctions (MTJ) is fully formed by stage 15/16 of embryonic development (approximately 13 hours AEL). (see Bate 1990, and Buttgereit et al 1996). Because the tendon cells must be resistant to the strong forces exerted by the attached muscles, the MTJ is characterized by extensive hemi-adherens junction contacts where both the muscle and the tendon attach to a prominent extracellular matrix (Tepass and Hartenstein 1994). Spontaneous muscle twitching begins in the embryo around stage 16 indicating that these muscles begin to be functional around this time.

These extensive electron dense adhesions are fully formed several hours before the dendrites of the multidendritic neurons have even begun to sprout their first dendrites. The growth of dendrites does not begin until 16 hours AEL at stage 16 (described in Gao et al 1999). Based on this well-described developmental sequence, it seems like it is already known that physical occlusion of apodeme territory prevents penetration of dendrites into this territory.

Recommend revising this statement to reflect this prior knowledge:

“These distinct orientations likely reflect physical occlusion of apodeme territory by muscles which attach to their tendon cells during embryonic development several hours prior to the sprouting of sensory neuron dendrites (Bate 1990, and Buttgereit et al 1996). However, experimental evidence supporting this idea is lacking. Furthermore, whether there are attractive cues at apodeme junctional domains, and/or the repulsive cues at other epidermal cell-cell interfaces is currently unknown”.

**Have all data underlying the figures and results presented in the manuscript been provided?**

Reviewer #1: Yes

Reviewer #2: Yes

Reviewer #3: Yes

PLOS authors have the option to publish the peer review history of their article (what does this mean?). If published, this will include your full peer review and any attached files.

Reviewer #1: **Yes: **Cheng-Ting Chien

Reviewer #2: No

Reviewer #3: No

---

## [Editor Report · Decision Letter 2]

29 Mar 2024

Dear Dr Parrish,

We are pleased to inform you that your manuscript entitled "Dendrite intercalation between epidermal cells tunes nociceptor sensitivity to mechanical stimuli in Drosophila larvae" has been editorially accepted for publication in PLOS Genetics. Congratulations!

Yours sincerely,

Fengwei Yu

Academic Editor

PLOS Genetics

Gregory P. Copenhaver

Editor-in-Chief

PLOS Genetics

Comments from the reviewers (if applicable):

**Data Deposition**

http://datadryad.org/submit?journalID=pgenetics&manu=PGENETICS-D-23-01043R2

**Press Queries**

---

## [Editor Report · Acceptance letter]

18 Apr 2024

PGENETICS-D-23-01043R2 

Dendrite intercalation between epidermal cells tunes nociceptor sensitivity to mechanical stimuli in Drosophila larvae 

Dear Dr Parrish, 

We are pleased to inform you that your manuscript entitled "Dendrite intercalation between epidermal cells tunes nociceptor sensitivity to mechanical stimuli in Drosophila larvae" has been formally accepted for publication in PLOS Genetics! Your manuscript is now with our production department and you will be notified of the publication date in due course.

With kind regards,

Lilla Horvath

PLOS Genetics

On behalf of:
